

# Mapping soil slaking index and assessing the impact of management in a mixed agricultural landscape

Edward J. Jones[1], Patrick Filippi[1], Rémi Wittig[2], Mario Fajardo[1], Vanessa Pino[1], and Alex. B. McBratney[1]

[1]School of Life and Environmental Sciences & Sydney Institute of Agriculture, Faculty of Science, The University of Sydney, New South Wales, Australia
[2]École Nationale Supérieure d'Agronomie et des Industries Alimentaires (ENSAIA), University of Lorraine, France.

**Correspondence:** Edward J. Jones (edward.jones@sydney.edu.au)

**Abstract.** Soil aggregate stability is a useful indicator of soil physical health and can be used to monitor condition through time. A novel method to quantify soil aggregate stability, based on the relative increase in the footprint area of aggregates as they disintegrate when immersed in water, has been developed and can be performed using a smartphone application - SLAKES. In this study the SLAKES application was used to obtain slaking index (SI) values of topsoil samples (0 to 10 cm) at 158 sites to

assess aggregate stability in a mixed agricultural landscape. A large range in SI values of 0 to 7.3 was observed. Soil properties and land use were found to be correlated with observed SI values. Soils with clay content >25% and CEC:clay ratio >0.5 had the highest observed SI values. Variation in SI for these soils was driven by OC content which fit a segmented exponential decay function. An OC threshold of 1.1% was observed below which the most extreme SI values were observed. Soils under dryland and irrigated cropping had lower OC content and higher observed SI values compared to soils under perennial cover.

These results suggest that farm managers can mitigate the effects of extreme slaking by implementing management practices to increase OC content, such as minimum tillage or cover-cropping. A regression-kriging method utilising a Cubist model with a suite of spatial covariates was used to map SI across the study area. Accurate predictions were produced with leave-one-out cross-validation (LOOCV) giving an LCCC of 0.85 and an RMSE of 1.1. Similar validation metrics were observed in an independent test set of samples consisting of 50 observations (LCCC = 0.82; RMSE = 1.1). The potential impact of implementing

management practices that promote soil OC sequestration on SI values in the study area was explored by simulating how a 1% increase in OC would impact SI values at observation points, and then mapping this across the study area. Overall, the maps produced in this study have the potential to guide management decisions by identifying areas that currently experience extreme slaking, and those areas that are expected to have a significant reduction in slaking by increasing OC content.



## 1   Introduction

Objective and quantitative metrics are required to assess soil health and monitor soil condition through time. Acquisition of such metrics should be low-cost, rapid and simple to facilitate sufficient spatial and temporal sampling density. Aggregate stability is an important indicator of physical condition that quantifies a soil's resistance to slaking and dispersion. Slaking is the disintegration of soil aggregates as a result of rapid wetting (Yoder, 1936; Oades and Waters, 1991). Slaking occurs when

soil aggregates are unable to withstand the stress induced by water uptake derived from two main causes: swelling of clay minerals as water is adsorbed into the interstitial space; and internal pressure caused by compression of entrapped air bubbles as capillary action draws water into the small pores between soil particles (Emerson, 1964). Most cultivated soils in Australia are prone to some degree of slaking. The degree of slaking determines if the process produces a favourable or unfavourable environment for cultivation and plant growth. A small degree of slaking can be beneficial and is associated with self-mulching

- an ability to recover from disturbance by reforming small (<5 mm) aggregates at the soil surface following wetting and drying cycles (Grant and Blackmore, 1991); and mellowing - a partial disintegration of soil aggregates on wetting that results in increased friability (Barzegar et al., 1996). Slaking produces detrimental effects when aggregates disintegrate further into microaggregates (ø <0.25 mm). Detached microaggregates migrate and settle into pores, reducing pore volume, decreasing infiltration and percolation rates, and leading to increased surface runoff (Rengasamy et al., 1984). Erosion susceptibility is

exacerbated as greater run-off volumes increase erosive power and the slaked aggregates also provide suitably sized particles for translocation. Ultimately the soil has a lowered capacity to support plant growth as plant available water and soil-atmosphere gas exchange are both reduced. In severe cases, crusting or hard-setting occurs when slaked and dispersed aggregates coalesce and set hard on drying (Mullins et al., 1990). Soil strength increases as the soil dries producing difficulty in cultivation until the soil is rewetted, and shoot emergence and root growth may be restricted (Mullins et al., 1990).

The susceptibility of a soil aggregate to slake is related to texture, mineral composition and organic matter content (Mullins et al., 1990). Soils with high clay content, especially those containing smectite or vermiculite minerals, are more likely to slake as they expand on wetting and also contain a greater number of small diameter pores into which capillary action will draw water and compress entrapped air-bubbles (Emerson, 1964). High organic matter content improves soil structure by binding soil particles into stable aggregates and reducing susceptibility to slaking (Chenu et al., 2000). Agricultural management

practices that increase susceptibility to slaking include: conventional tillage methods that destroy soil structure and accelerate organic matter decomposition; burning or removal of crop residues; and the application of pesticides and other chemicals that are harmful to soil biota and lead to disruption of organic matter cycling and reduced aggregation. The detrimental effects of soil slaking are more pronounced in areas with clear wetting and drying cycles, such as temperate Australia, as the initial water content of soil affects the degree of slaking upon rewetting (Collis-George and Lal, 1971).

Slaking and dispersion are quantified through aggregate stability tests that observe changes in soil aggregate morphology following immersion in water in an attempt to predict soil behaviour in the field. Emerson (1967) developed a test to classify samples into eight classes based on the degree of slaking, swelling and dispersion observed when air-dried soil aggregates are immersed in distilled water. The Emerson Aggregate Test was extended by including a supplementary analysis whereby soil




samples were wetted and moulded into cubes before immersion in the distilled water as a means to simulate the shear forces

associated with raindrop impact and tillage on bare soil (Loveday and Pyle, 1973; Emerson, 1991). Field et al. (1997) modified these tests further to include observations of slaking and dispersion at both ten minutes and two hours post submersion in the 'aggregate stability in water' (ASWAT) test. This greatly decreased the time-requirement from 20+ h required for previous tests, however, interpretation of the degree of slaking for the ASWAT test remained moderately subjective and scores were produced on an ordinal scale from 0 to 4 which limits statistical applications.

A new method has been developed to calculate degree of slaking using a time-series of digital photographs to quantify the increase of the footprint area of aggregates as they disintegrate when immersed in distilled water (Fajardo et al., 2016). This method has been incorporated into a smartphone application, SLAKES, that is able to quantify aggregate stability in only ten minutes (Fajardo and McBratney, 2019). The reduced assessment time was achieved as the authors found that the two hour reading can be reliably estimated from change in footprint area over the ten minute analysis period. The SLAKES application

requires no specialty equipment and the automated nature of the application allows aggregate stability to be quantified with minimum training. These advances make the analysis more readily available to farm managers and citizen scientists. The method calculates an objective and continuous slaking index (SI), which reduces operator error and facilitates elucidation of contributing factors of observed slaking. For example, Flynn et al. (2020) investigated aggregate stability of Vertisols under different agricultural management strategies and found that SI was significantly more sensitive at distinguishing the perennial,

no-till and conventional tillage management treatments compared to the Cornell Wet Aggregate Stability Test (Schindelbeck et al., 2016).

Few studies have mapped aspects of soil aggregate stability using digital soil mapping (DSM) techniques. Odeh and Onus (2008) used regression-kriging and indicator-kriging to model the electrochemical stability index (ESI) across an irrigated cropping region of western NSW, Australia. This resulted in a map of 'risks zones' that were susceptible to dispersion and

which could be prioritised for increased monitoring and tactical management to abate immediate and future detrimental impacts on crop production. A study by (Annabi et al., 2017) also utilised regression-kriging to produce accurate predictions of soil aggregate stability of an agricultural district in Tunisia. Fine-resolution maps of soil aggregate stability across fields and farms have considerable potential to aid farm managers in decision-making processes. Such maps could help guide farm managers to implement soil amelioration practices, such as tactical application of gypsum, or change in management practices, such as

cultivation method or use of cover-crops.

The current study investigated the use of the SLAKES application and DSM techniques to assess variation in SI across a landscape with different agricultural and natural land uses. The contribution of both soil attributes and land management to slaking was explored, and the potential impact of increasing soil OC levels on slaking was explored.



## 2 Methodology

### 2.1 Site description


The study was centred around a mixed farming property, L'lara (30°15'18" S, 149°51'39" E), which is located ~11 km north-east of the township of Narrabri, NSW, Australia (Fig. 1). Climate at the study site is classified as humid subtropical (Cfa) under the Köppen-Geiger system (Peel et al., 2007). The site experiences hot summers and cool winters. The long-term average annual precipitation for the study area is 658 mm, and is slightly summer-dominant (Bureau of Meteorology, 2020).

The landscape at L'lara and its surrounds can be broadly characterised into two distinct areas: sand covered hills derived predominantly from Jurassic coarse-grained sediments of Pilliga sandstone covered by Quaternary sands and talus material; and floodplain areas derived from Quaternary alluvial deposits of basaltic materials washed from the western side of the Nandewar range. The soils of the floodplain area at L'lara are classified as black and brown Vertosols according to the Australia Soil Classification (Isbell, 2016), with small areas of grey Vertosols. The soils on the sand hill area were predominately Chromosols,

Dermosols, Kandosols, Rudosols and Tenosols (Isbell, 2016), the unifying feature of these soils was the presence of a relatively sandy topsoil. L'lara encompasses a total area of 1,850 ha, with approximately 1,070 ha used for dryland, broadacre cropping. Cropping is performed primarily on the Vertosols, and occurs over both summer and winter periods with cotton (*Gossypium hirsutum* L.), wheat (*Triticum aestivum* L.), canola (*Brassica napus* L.) and chickpea (*Cicer arietinum* L.) grown in rotation. Lower lying floodplain areas close to creek lines and all of the sand hill area is used for grazing of beef cattle on unimproved

native pastures (~704 ha) and remnant forest cover (~76 ha).

L'lara lies at the centre of a diverse landscape. Outside the property, dryland cropping and grazing occur on the floodplains and slopes to the east and south. Intensive irrigated agricultural production occurs on the lower floodplain to the south-west of the property, and the Killarney State Conservation Area lies directly to the north. This conservation area contains similar species as the remnant forest area found on L'lara which is dominated by white cypress pine (*Callitris glaucophylla*), hickory

(*Acacia leiocalyx*), black cypress pine (*Callitris endlicheri*), narrow-leaved ironbark (*Eucalyptus crebra*), bulloak (*Allocasuarina leuhmannii*) and dirty gum (*Eucalyptus chloroclada*).

### 2.2 Soil sampling

Sample sites were identified on L'lara and the surrounding area. The majority of on-farm samples (n = 58) were identified based on a random stratified sampling approach utilising soil type and land use as parameters (Fig. 1). This ensured representation of

the major soil types and different land uses - dryland cropping, pasture and forest cover found on the property. To investigate slaking in the area surrounding L'lara, an additional 50 samples were sourced from neighbouring properties found within a 5 km distance from the boundary of L'lara. These sites were identified through a random stratified approach utilising elevation, MrVBF and airborne gamma radiometrics as input variables (Filippi et al., In Press). K-means clustering was utilised to split the data into four classes roughly equivalent to sand hill, transition, upper floodplain and lower floodplain landscape positions.

Sample sites were randomly selected within each stratum. Five of the samples on the lower floodplain were under irrigated agriculture, a land use not represented on L'lara. A supplementary dataset of 30 existing sites on the dryland cropping areas



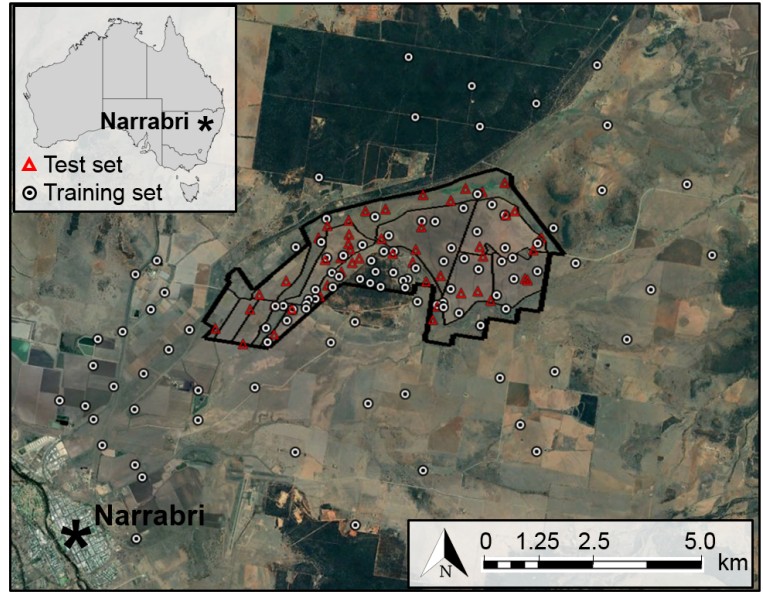

**Figure 1.** Location of L'lara farm and the wider study area in relation to the township of Narrabri, NSW, Australia. Sample locations used as a training set (n = 108) and test set (n = 50) are indicated. Satellite imagery sourced from Google Earth Pro V 7.3.2.5776. (March 5, 2019). Narrabri, NSW, Australia. 30° 16' 31.37"S, 149° 51' 46.42"E, Eye alt 20.57 km. Image © CNES/Airbus 2020. http://www.earth.google.com [April 20, 2020].

of L'lara which are described in Filippi et al. (2019), and 20 sites on the pasture areas were also used as a test set for model predictions. At each of the 158 sites a topsoil (0 to 10 cm) sample was obtained by excavation using a shovel at a discrete location.

## 2.3 Laboratory methods

All soil samples were air-dried at 40°C for 48 hours. A selection of 20 to 30 aggregates were isolated and retained after drying for use with the SLAKES application. The remaining sample was then ground to pass through a 2 mm sieve prior to laboratory analysis. Particle size analysis was performed using the hydrometer method (Gee and Bauder, 1986). Organic carbon content was quantified using the Walkley-Black method (Walkley and Black, 1934). Soil pH and electrical conductivity (EC) was measured using a 1:5 soil:$H_2O$ suspension. As the soil samples did not contain significant quantities of carbonates or soluble salts, the cation exchange capacity (CEC) was assessed using the ammonium acetate method (Rayment and Lyons, 2011). Exchangeable sodium percentage (ESP) and Ca:Mg ratio were calculated from the relevant exchangeable cations, and CEC:clay ratio was calculated following correction for the CEC contribution of organic matter (OM).



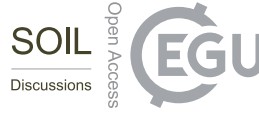

## 2.4 SLAKES slaking index

A selection of 20 to 30 soil aggregates (ø 5-10 mm) were isolated from the air-dried, bulk soil samples prior to grinding and sieving for laboratory analysis. If distinct aggregates were not immediately evident in the bulk soil then the sample was passed through a 5 mm sieve to isolate aggregates, this procedure was often required in sandy soils. A slaking index (SI) was obtained using the SLAKES application (Fajardo and McBratney, 2019). Briefly, a smartphone (Galaxy J2 Pro, Samsung, Republic of Korea) with an 8 MP digital camera was fixed on an articulated stand to provide the camera lens an unimpeded view of the

bench surface. The height of the stand was adjusted so that the field of view of the camera was filled by a 100 mm diameter petri dish placed on the surface of the bench directly below the camera. Three soil aggregates were placed into the petri dish and an initial image of the aggregates was acquired. The petri dish was then drawn back and replaced by an identical petri dish filled with sufficient deionised water to completely immerse the aggregates. The aggregates were held directly above the deionised water and dropped simultaneously into the petri dish with care being taken to preserve the order and orientation of

the aggregates to that of the initial image. The start button of the SLAKES application was then immediately pressed and the setup left to process over a 10 minute period after which the SI was displayed on the screen of the smartphone. The experiment was performed on a white surface to increase contrast between the soil aggregates and the background surface. The experiment was also performed under diffuse and constant lighting to prevent the occurrence of shadows over the petri dish which could introduce errors during the image segmentation process. The procedure was repeated twice for each sample and if the difference

between the duplicate readings was greater than one unit an additional reading was obtained. Outlier readings were discarded and replicate readings averaged to provide the final SI for each sample.

The SLAKES application uses an image segmentation approach to calculate the footprint area of each aggregate, expressed as pixel count, and tracks the relative increase in area of individual aggregates as they breakdown over time (Fajardo et al., 2016). The SI of an individual aggregate at a given time after immersion is calculated as:

$$SI_t = \frac{A_t - A_0}{A_0} \tag{1}$$

where, the $A_0$ is the initial footprint area of the aggregate and $A_t$ is the footprint area if the aggregate at time, $t$. An SI of 0 means that the footprint area of the aggregate has not increased at all, an SI of 1 means that the footprint area has increased in size by 100%, an SI of 2 means that the footprint area has increased in size by 200%, etc. The change in SI over the course of the analysis is used to fit a Gompertz function on a log time-scale and calculate parameters $a$, $b$, $c$:

$$SI_t = ae^{-be^{-c \cdot log(t)}} \tag{2}$$

where, as described in Fajardo et al. (2016), $a$ is an asymptote representing the maximum SI after an indefinite period of time, $b$ describes displacement along the time axis and is associated with initial slaking, $c$ describes the growth rate and is associated with ongoing slaking of the aggregate. The SI value returned from the SLAKES application is the average of the $a$ parameter calculated individually for each aggregate. A major benefit of this approach is that this value can be estimated after only 10

minutes of immersion, unlike other approaches that require $\geq 2$ hours.





## 2.5 Spatial covariates for modelling and mapping slaking index

A range of publicly available spatial datasets were used as input variables to model SI across the study area (Table 1). This included satellite imagery, a digital elevation model, terrain attributes, air-borne $\gamma$-radiometric maps, and a lithology indicator. Landsat 7 tier 1 surface reflectance satellite imagery from 2000 to 2018 was accessed through Google Earth Engine (Gorelick et al., 2017). To remove pixels that were affected by cloud cover or shading, a cloud-masking filter was applied to all images. The Normalized Difference Vegetation Index (NDVI) was then calculated for each pixel in each image. The 5th, 50th and 95th percentile of the time-series of NDVI values were then determined for each pixel. The reason for using different NDVI percentiles was to characterise spatial variability in vegetation cover and vigour over the nineteen year period. For example, the median (50th percentile) gives a value of typical greenness and the 95th percentile gives peak plant greenness. The 5th percentile would likely be low and represent soil variability for areas that are tilled or heavily grazed, and remain higher for areas of perennial cover such as forests.

**Table 1.** Description and source of covariates used for digital soil mapping.

| Type | Description | Resolution | Source |
|---|---|---|---|
| Satellite imagery[†] | Landsat 7 NDVI 5% | 30 m | Google Earth Engine |
| | Landsat 7 NDVI 50% | 30 m | Google Earth Engine |
| | Landsat 7 NDVI 95% | 30 m | Google Earth Engine |
| Terrain | DEM (m) | 5 m | NSW Government |
| | Slope (%) | 30 m | CSIRO |
| | Aspect | 30 m | CSIRO |
| | MrVBF | 30 m | CSIRO |
| | MrRTF | 30 m | CSIRO |
| $\gamma$-radiometrics | Total dose | 100 m | Geoscience Australia |
| | Potassium (%) | 100 m | Geoscience Australia |
| Lithology | Silica (%) | ~125 m | Gray et al. (2016) |

[†] Landsat 7 NDVI values represent percentiles computed over the 2000 to 2018 time-period

A 5 m digital elevation model (DEM) was accessed through the ELVIS (ELeVation Information System) platform (ANZLIC, 2019). This DEM was derived from photogrammetry and generated via airborne imagery. Shuttle Radar Topography Mission (SRTM) derived terrain attributes at 30 m resolution were also accessed through CSIRO's Data Access Portal (CSIRO, 2019). The specific terrain attributes obtained included aspect, multi-resolution ridge-top flatness (MrRTF), multi-resolution valley bottom flatness (MrVBF), and slope. Gridded gamma radiometric data at 100 m spatial resolution derived from an air-borne gamma ray spectrometer was obtained through the Geophysical Archive Data Delivery System (GADDS) (Geoscience Australia, 2019). The individual datasets used included dose rate, and potassium concentration data, which were processed with low-pass filtering (Minty et al., 2009). A map of silica index, which is essentially a map of silica content of soil parent mate-





rial, was also used as a covariate (Gray et al., 2016). The silica index is known to relate to soil texture and other important soil physical properties, such as water holding capacity.

## 2.6 Modelling and mapping procedure

A regression-kriging approach was utilised to map SI across the study area. All data handling and processing was performed in the open software R (R Core Team, 2019). The data set was split into a training set (n = 108) and a test set (n = 50). At each of

the 108 sampling sites in the training set, the spatial covariates described in Table 1 were extracted using the nearest neighbour method. A Cubist model was then used to build a relationship between SI and the spatial covariates at each observation point (Kuhn and Quinlan, 2020). A 20 m grid of the study area was created and the spatial covariates were then extracted using the nearest neighbour method at each grid point. The developed Cubist model was then used to predict SI on this grid of the study area. The residuals (difference between the observed and predicted SI values) at observation points were kriged onto the

same 20 m grid to account for spatial auto-correlation of residuals. The kriged residuals was added to the mapped output of the Cubist model to obtain the final SI prediction map of the study area. The complexity of the Cubist model was fine-tuned using a leave-one-out cross-validation (LOOCV) approach on the training set. The external validation test set consisting of 50 sites was used to assess the final model. Validation metrics used to assess the prediction performance were the Lin's concordance correlation coefficient (LCCC), root-mean-square error (RMSE), bias and the coefficient of determination ($R^2$).

## 2.7 Mapping the simulated effect of increased soil organic carbon on slaking index

Options to reduce soil slaking were investigated as a means to inform management practices. To achieve this the relationship between SI and measured soil properties was explored to identify potential causal factors of slaking and allocate observation points into classes with similar behaviour. Class-based regression was then used to construct individual predictive models between SI and these other measured soil attributes using either multiple linear regression or segmented, non-linear regression

for more complex relationships (Baty et al., 2015). The effect of increasing soil OC levels on SI investigated by simulating a 1% increase in OC and applying the relevant class-based regression equation using the laboratory data at each point. These modified SI values were then extrapolated across the study area using the same regression-kriging approach as described above and validated using a LOOCV approach.

## 3 Results and discussion

## 3.1 Investigating slaking index variation

### 3.1.1 Slaking index and soil properties

A large range in SI was observed for the samples analysed in this study (Table 2). A minimum SI of 0 was observed for nine samples, meaning that no slaking or swelling occurred and the footprint area of these soil aggregates did not increase. A maximum SI of 7.3 was observed, meaning that the average footprint area for these aggregates increased in size by 730%. This



indicates an extreme level of aggregate disintegration, although it remains below the maximum theoretical SI of 7.8 suggested by Fajardo et al. (2016). Organic C had an observed range of 0.33 to 2.97% and a median value of 0.88%, demonstrating that many of the sampled locations had low levels of OC. Other measured soil properties ranged widely, demonstrating the diversity of soils sampled, e.g. clay ranged from 2.5 to 60.2% and pH ranged from 4.8 to 9.2.

**Table 2.** Summary statistics of slaking index and laboratory derived soil properties.

| Property | Min. | 1st Qu. | Median | Mean | 3rd Qu. | Max. |
|---|---|---|---|---|---|---|
| Slaking index | 0.0 | 0.4 | 2.6 | 2.7 | 4.8 | 7.3 |
| Organic carbon (%) | 0.33 | 0.74 | 0.88 | 1.07 | 1.22 | 2.97 |
| Clay (%) | 2.5 | 11.1 | 29.1 | 28.1 | 42.1 | 60.2 |
| pH(1:5 H$_2$0) | 4.8 | 6.0 | 6.8 | 7.0 | 8.3 | 9.2 |
| EC (dS m$^{-1}$) | 0.01 | 0.04 | 0.12 | 0.15 | 0.19 | 0.81 |
| Exch. Ca$^{+2}$ (cmol$_c$ kg$^{-1}$) | 0.0 | 1.7 | 10.5 | 11.3 | 19.8 | 34.0 |
| Exch. Mg$^{+2}$ (cmol$_c$ kg$^{-1}$) | 0.0 | 0.7 | 5.1 | 6.0 | 11.0 | 17.0 |
| Exch. K$^+$ (cmol$_c$ kg$^{-1}$) | 0.1 | 0.4 | 0.8 | 0.9 | 1.4 | 2.2 |
| Exch. Na$^+$ (cmol$_c$ kg$^{-1}$) | 0.0 | 0.0 | 0.2 | 0.5 | 0.6 | 3.6 |
| CEC (cmol$_c$ kg$^{-1}$) | 0.2 | 2.8 | 15.6 | 18.8 | 32.8 | 52.8 |
| ESP (%) | 0.2 | 1.0 | 1.8 | 2.9 | 3.6 | 19.4 |
| Ca:Mg ratio | 0.1 | 1.5 | 1.9 | 2.2 | 2.5 | 10.8 |
| CEC:clay ratio | 0.01 | 0.08 | 0.44 | 0.43 | 0.70 | 1.09 |

   Slaking index (SI) was positively correlated with clay content (r = 0.84), pH (r = 0.70), electrical conductivity (r = 0.44),
CEC (r = 0.87), CEC:clay ratio (r = 0.84) and all exchangeable cations (Table 3). Weak negative correlations were observed for SI with OC (r = -0.31) and Ca:Mg (r = -0.26). These observations support Fajardo et al. (2016) findings that SI was positively correlated with pH, clay content and exchangeable Na$^+$ and Mg$^{+2}$, and negatively correlated with Ca:Mg. The strongest correlation with SI in this study was observed with exchangeable Mg$^{+2}$ (r = 0.90). This is in contrast to a recent study that demonstrated exchangeable Mg$^{+2}$ played a negligible role in flocculation of soil particles and aggregate stability (Zhu
et al., 2019). It is believed that the observed correlation in our study is due to the dependence of exchangeable Mg$^{+2}$ on clay content, CEC and shrink-swell minerals, such as smectite, rather than a direct causal effect. Clay content was a strong indicator of SI potential. Only one sample with clay content <25% had an observed SI greater than 1, in contrast only three samples with clay content ≥ 25% had an observed SI less than 1. Clay soils are often more susceptible to slaking as they have both a higher concentration of shrink-swell minerals and also a greater concentration of smaller pores that may trap and compress air
bubbles (Emerson, 1964). The high CEC:clay ratios observed in these samples and correlation with clay content indicate that the dominant phyllosilicate in clay soils studied is smectite. No correlation was observed between SI and ESP in our study. Churchman et al. (1993) reviewed causes of swelling and dispersion in Australian soils and identified that exchangeable Na$^+$ increased swelling, but only for high ESP values. Most of the samples in our study had low ESP values, which explains the





lack of correlation with SI values. The low ESP values resulted in minimal dispersion observed in these samples, which was
beneficial for this study as the SLAKES application currently cannot distinguish between slaking and dispersion (Fajardo et al.,
2016).

**Table 3.** Pearson correlation coefficient (r) between soil properties.

| | SI | OC | Clay | pH | EC | Exch. Ca | Exch. Mg | Exch. K | Exch. Na | CEC | ESP | Ca:Mg |
|---|---|---|---|---|---|---|---|---|---|---|---|---|
| OC | -0.31 | | | | | | | | | | | |
| Clay | 0.84 | -0.13 | | | | | | | | | | |
| pH | 0.70 | -0.20 | 0.85 | | | | | | | | | |
| EC | 0.44 | 0.07 | 0.58 | 0.47 | | | | | | | | |
| Exch. Ca | 0.83 | -0.15 | 0.84 | 0.83 | 0.45 | | | | | | | |
| Exch. Mg | 0.90 | -0.22 | 0.85 | 0.74 | 0.45 | 0.92 | | | | | | |
| Exch. K | 0.59 | 0.19 | 0.65 | 0.60 | 0.63 | 0.67 | 0.65 | | | | | |
| Exch. Na | 0.64 | -0.25 | 0.68 | 0.65 | 0.63 | 0.60 | 0.65 | 0.52 | | | | |
| CEC | 0.87 | -0.17 | 0.87 | 0.82 | 0.49 | 0.99 | 0.97 | 0.70 | 0.66 | | | |
| ESP | 0.01 | 0.04 | 0.01 | -0.10 | 0.16 | -0.09 | -0.04 | -0.10 | 0.40 | -0.06 | | |
| Ca:Mg | -0.26 | 0.22 | -0.20 | -0.04 | -0.07 | -0.10 | -0.27 | -0.04 | -0.18 | -0.16 | -0.24 | |
| CEC:clay | 0.84 | -0.21 | 0.74 | 0.72 | 0.38 | 0.94 | 0.91 | 0.62 | 0.53 | 0.94 | -0.14 | -0.16 |

SI, slaking index; OC, organic carbon; pH, pH(1:5 $H_2O$); EC, electrical conductivity (1:5 $H_2O$); CEC, cation exchange capacity; ESP, exchangeable sodium percentage; Ca:Mg, ratio of exchangeable $Ca^{+2}$ to $Mg^{+2}$; CEC:clay, ratio of organic matter corrected CEC to clay content.

### 3.1.2 Slaking index and land use

Land use at sampling sites were categorised into four classes: forest, predominately remnant vegetation cover on sand hills;
pasture, encompassing improved/unimproved pastures but also stock routes and other areas of perennial grass cover; dryland
cropping; and irrigated cropping. Clear differences in SI values were observed under these different land uses, which were
accentuated after separating based on clay content (Fig. 2). For samples with clay content $\geq 25\%$, irrigated cropping had the
highest SI values, followed by dryland cropping (which showed a large range of SI values), and then pasture. No samples
with clay content $\geq 25\%$ were observed under forest cover, nor soils with clay content <25% under irrigated cropping. These
findings are supported by the few existing studies investigating SI values of aggregates under cultivated sites compared to
paired sites under natural vegetation Fajardo et al. (2016); Flynn et al. (2020). Decreased aggregate stability of soils under
cropping compared to pasture or natural vegetation has also been observed by other indicators of aggregate stability, such
as mean weight diameter and water stable aggregates (Saygın et al., 2012; Ye et al., 2018). The marked differences in soil
aggregate stability between land uses may be attributable to the impact of cultivation on the soil - both the direct destruction of
aggregates through cultivation and associated increase in soil respiration and loss of OC. In a review of The natural disposition
of these soils to slake is evident with an average SI of 2.8 observed for soils with $\geq 25\%$ clay content under perennial ground
cover in the pasture land use. This natural disposition had been exacerbated by cultivation with an average SI value of 4.8




observed for sites under dryland cropping, and 5.0 for sites under irrigation. The higher level of slaking under irrigation may

be due to the fact that irrigated cropping represents a further level of cultivation intensification compared to dryland sites and

sampled irrigated sites also only occurred on soils with clay content >50%. For those sites with clay content <25% SI values

were predominately <1. Differences between land use were not as distinct for these low clay content soils although increases

in mean values were observed from forest to pasture and then dryland agriculture. A wide range of SI values was observed for

samples with $\geq 25\%$ clay content, warranting further investigation.

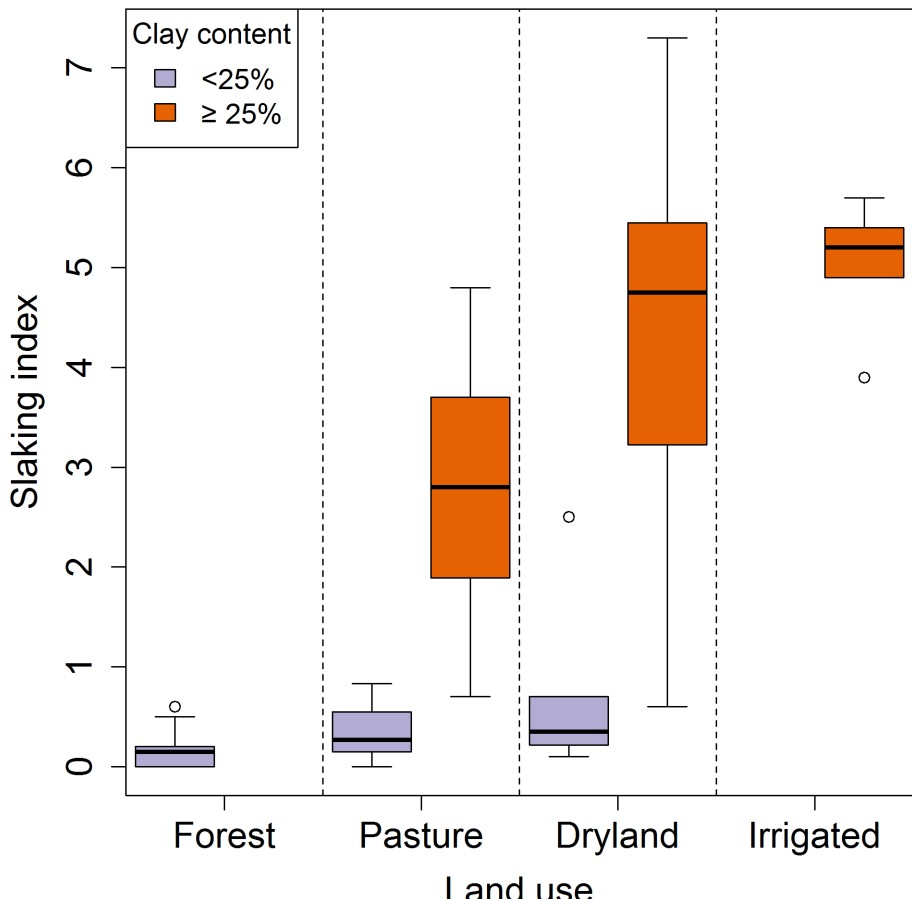

**Figure 2.** Boxplots of slaking index grouped by land use (forest, pasture, dryland cropping or irrigated cropping) and clay content (<25% or $\geq 25\%$).

### 3.1.3    Effect of organic carbon on slaking index

Organic C has been shown to increase soil aggregation and decrease susceptibility to slaking (Six et al., 2000). Chenu et al.

(2000) found OC to be a good predictor of soil aggregate stability ($R^2 = 0.72$) when investigating the effects of tillage manage-

ment on humic loamy soils in southwest France. The diverse range of soils used in this study are assumed to have confounded




this relationship as only a weak negative correlation (r = -0.31) between SI and OC was observed while much stronger correlations were observed for other soil properties such as clay content or CEC:clay ratio (Table 3). To investigate these correlations further, the relationship between clay content, CEC:clay ratio and SI was visualised (Fig. 3). CEC:clay ratio was chosen as an

important parameter as it is a useful indicator of clay mineral type which affects slaking through contribution to the shrink-swell characteristics of a soil. A correlation between clay content and CEC:clay ratio was observed. This relationship was related to landscape position in the study area, as high clay content soils found on floodplain areas also contained a higher proportion of shrink-swell clay minerals, such as smectite. Meanwhile, topsoil samples from the hills and slopes had lower clay content and also a lower CEC:clay ratio, indicating the dominance of low CEC phyllosilicates, such as kaolinite or illite.

As identified previously, samples with clay content <25% showed minimal slaking. For samples with a clay content $\geq 25\%$, CEC:clay ratio was an important predictor of slaking. For example, soils with a clay content ~40% showed low to moderate slaking for CEC:clay ratio <0.5 and moderate to extreme levels of slaking for CEC:clay ratio >0.5 3). Clear threshold values were observed with extreme slaking values only occurring for soils clay content $\geq 25\%$ and CEC:clay ratio >0.5. This observation was used to allocate samples into two classes: samples with clay content $\geq 25\%$ and CEC:clay ratio >0.5; and all

remaining sample. Relationships between measured soil properties and observed SI values were modelled independently for each class as different critical values were expected to control behaviour of different soil classes (Loveland and Webb, 2003).

Soil organic carbon was the only significant predictor of SI for soils with clay content $\geq 25\%$ and CEC:clay ratio >0.5. The relationship between SI and OC fit a segmented, exponential decay function (Fig. 4). This equation was developed by optimising a four parameter nonlinear regression model to minimise residual sum of squares using the *nls* functions from

the *nlstools* R package (Baty et al., 2015). The model contained: a constant value that characterised SI behaviour under low OC levels; a threshold value above which the relationship was characterised by exponential decay; and two parameters that characterised exponential decay behaviour at high OC levels. A threshold value of 1.1% OC was identified. The average observed SI values for samples below this threshold was 5.01 - the highest observed SI values in this study. Extreme SI values were uniquely observed for samples with OC content under this threshold value. As the constant value indicates, no relationship

between OC and SI was identified for these samples, nor could a relationship between SI and other measured soil properties be identified. As such the factors responsible for the large range in observed SI values for these soils remains unidentified. To identify causal factors future research should investigate potential relationships between SI and OC fractions, OC type, microbial activity or crop species that have been previously been identified as influencing aggregate stability (Six et al., 1998; Morel et al., 1991; Blankinship et al., 2016). The 1.1% threshold value also effectively separated observed differences in OC

content between pasture and cropping land use activities. Interestingly, pasture sites with ~1% OC had lower observed SI values than corresponding dryland agriculture sites indicating that direct effects of cultivation, extended fallow or monoculture production may influence observed SI values although the number of samples is too few for statistical analysis. Similar critical OC content values ranging from 1.1 to 2% have been identified when considering a soil's ability to provide nutrients for crop growth, or support microbial diversity (Aune and Lal, 1997; Zvomuya et al., 2008; Yan et al., 2000). For this study the 1.1%

OC value should not be interpreted as a target value for farm managers to achieve but rather it describes an absolute minimum threshold below which slaking is unpredictable and can result in extreme values. To abate potentially detrimental effects of

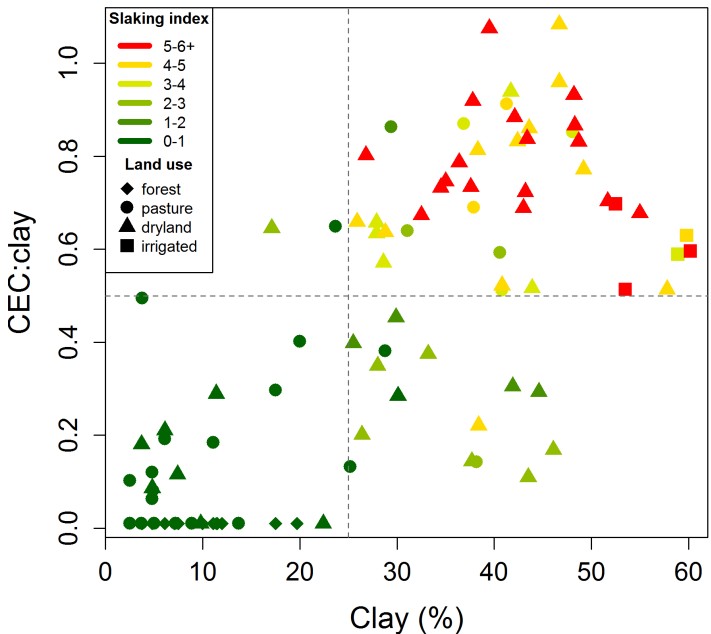

**Figure 3.** Relationship between clay content, CEC:clay ratio and slaking index (SI). Land use at sample site is indicated as either forest, pasture, dryland cropping or irrigated cropping. Dashed lines indicate clay content of 25% and CEC:clay ratio of 0.5 above which extreme slaking was observed.

slaking farm managers should aim to increase OC levels above this minimum threshold. The exponential decay component of the equation ($R^2$ = 0.27) suggests that slaking can be reduced, but not completely eliminated, by increasing OC content for the range of OC contents observed in this study. The constant parameter of 2.76 in the exponential decay function suggests a

minimum obtainable SI value for these soils, however this model was based on few observations and limited samples of >2% OC. Future investigation should prioritise identification of sites with higher OC content to better characterise this relationship.

The relationship between SI and OC for those soils that did not meet the criteria of $\geq 25\%$ clay content and CEC:clay ratio >0.5 were modelled separately using multiple-linear regression. For these soils, SI was explained with the following equation: SI = -0.22 - 0.19*OC + 0.09*clay ($R^2$= 0.77, RSE = 0.7, p = 0.000). This result demonstrates that while OC content still had

a significant effect on observed SI values, the relative effect was smaller for these soils. For example, soils with clay content $\geq 25\%$ and CEC:clay ratio >0.5 are expected to see a reduction in SI of 1.59 units if OC is increased from 0.7% to 1.7%, meanwhile if OC is increased from 0.7% to 1.7% in other soils a decrease in SI of only 0.19 is expected to occur. These two equations were used to model the effect on SI of a simulated 1% increase in OC at the sample sites, which was then mapped across the study area. The results of this analyses are shown in Section 3.2.4.

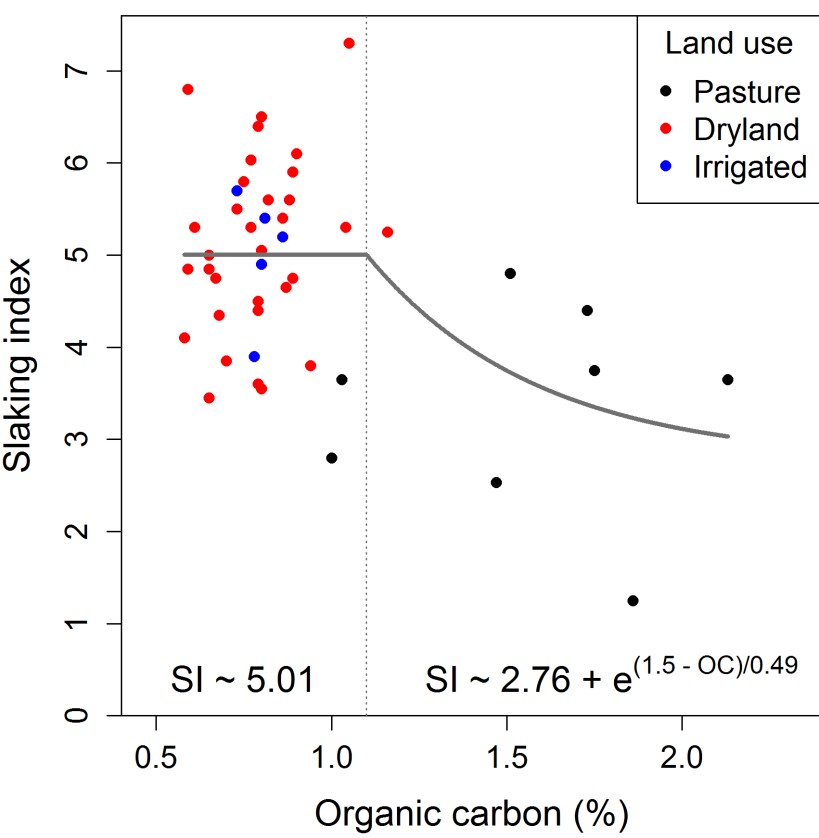

**Figure 4.** Relationship between slaking index (SI) and organic carbon (OC) for soil samples with clay content $\geq 25\%$ and CEC:clay ratio >0.5. A segmented, exponential decay function containing a lag phase and threshold value of 1.1% OC was fit to the observed data points. Land use at each observation point is indicated.

## 3.2 Mapping results

### 3.2.1 Importance of predictor variables

Investigation of the use of covariates as conditions and predictors in the Cubist model showed that MrVBF and the NDVI $5^{th}$, $50^{th}$ and $95^{th}$ percentiles were the most important predictor variables of SI values. The NDVI data used in this study largely represent variation in vegetation cover, and hence land use. The $5^{th}$ percentile NDVI was used as both a condition and a predictor in the model. The $5^{th}$ percentile NDVI represents the lower distribution of vegetation over the 2000-2018 period with low values indicating cultivated sites (Chen et al., 2018), and variation within cultivated sites representing topsoil varability. Low values $5^{th}$ percentile NDVI indicate areas of bare-earth from cultivation or extended fallow, facilitating the identification of cropping sites. For cropping sites, the $5^{th}$, $50^{th}$, and $95^{th}$ percentile values would be vastly different due to the seasonal nature





of cropping. This would be similar in the pastures due to seasonal 'browning off' in the perennial grass cover. In contrast, the
different NDVI percentiles for forest cover would be high and relatively similar due to the more constant biomass throughout
different seasons. The importance of NDVI percentiles in the model, and known relationships with land use support previous
findings that land use has a considerable influence on observed SI values (Fig. 2). The importance of MrVBF may be attributed
to the information it contains on landscape position, which is related to clay content and CEC:clay ratio (Gallant and Dowling,
2003). The lowest MrVBF values were found on the sand hills, increasing through a transition zone to the upper floodplain.
The highest MrVBF values were found on the lower floodplain, which also corresponded to the highest clay content in the
study area. Slope and gamma radiometric potassium data were used as predictors for some models. The important predictors
in the model reflect those used by Ye et al. (2018) to map aggregate stability in a small catchment of the Loess Plateau which
the authors found was explained by intrinsic factors (parent material, terrain attributes and soil type) and extrinsic factors (land
use and farming practice). The covariates that were the least important predictors included elevation, MrRTF and aspect.

### 3.2.2 Mapping accuracy

The quality of the predictions of SI from the regression-kriging approach was assessed using two validation techniques. The
first technique involved using LOOCV on the training dataset (n = 108). This method showed that SI could be predicted to
a relatively high degree of accuracy, with an LCCC of 0.85, $R^2$ of 0.75, RMSE of 1.1 and a bias of 0.0 (Fig. 5). The second
approach involved comparing SI values observed for an independent test set (n = 50) with SI values extracted from the final
map product. The second approach demonstrated the robustness of the model, as SI was predicted with similar accuracy to
that of the training set, with an LCCC of 0.82, $R^2$ of 0.78, RMSE of 1.1 and a bias of 0.6 (Fig. 5). This demonstrates that SI
can be accurately spatially predicted when using DSM techniques and ancillary spatial information. The successful prediction
of SI can be attributed to availability of ancillary spatial information that explain the main factors controlling slaking, such as
the different NDVI percentiles representing land cover and use, and MrVBF representing clay content and the accumulation of
water/soil. While there are no other published studies to our knowledge that have modelled and mapped SI across a study area,
these validation statistics are comparable to other DSM studies that have modelled other aspects of soil stability such as Annabi
et al. (2017) who modelled aggregate stability using three different indices in a study region in Tunisia, with an accuracy of
0.62 to 0.74 $R^2$ when tested with LOOCV.

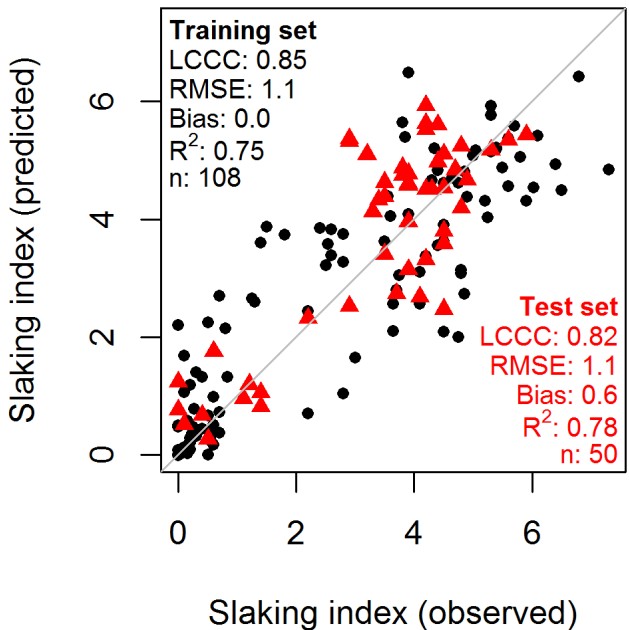

**Figure 5.** Plot of observed and predicted slaking index (SI) values from regression-kriging for two validation methods: (1) leave-one-out cross validation (LOOCV) on the training set (n = 108), and (2) external validation on an independent test set (n = 50).

### 3.2.3 Spatial variability of slaking index

340 The map of soil SI across the study area shows considerable variation (Fig. 6). The model was very effective at mapping high clay content soils that had a natural tendency to slake and also at identifying tillage practices that exacerbated this effect. It is clear that SI values were higher on arable areas, particularly on the cropped fields at L'lara, as well as the dryland and irrigated cropping areas lower down the floodplain to the south-west of L'lara. The forested areas showed the lowest SI values in the study area. The spatial patterns of the maps are clearly driven by vegetation cover/land use, and MrVBF, as indicated by the

345 variables used as conditions and predictors in the Cubist model. The unique patterns of MrVBF can be seen, as low SI values are found where deposition would be low, whereas high SI values are found where deposition would expected to be high. The NDVI 5th percentile covariate provides a good indication of whether a field has undergone tillage or been left in a bare fallow but provides no insight into the frequency, timing or intensity of tillage events. An aspect for further improvement to this approach would be to include a more sensitive method able to characterise the frequency of tillage events or quantify the

350 amount of time left under bare fallow.





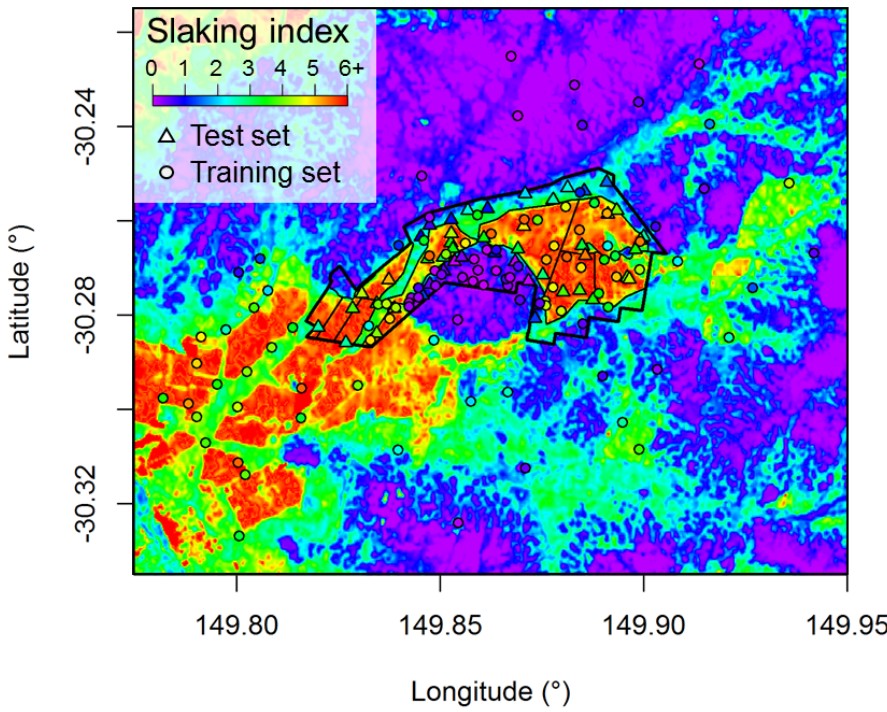

**Figure 6.** Prediction of slaking index (SI) across the study area using regression-kriging. The boundaries of L'lara farm are indicated as well as SI values at observation sites for the training set (n = 108) and test set (n = 50).

### 3.2.4 Mapping change in slaking index for a 1% increase in organic carbon

The impact of increasing soil OC levels by 1% on SI values was assessed and mapped across the study area (Fig. 7). When tested with LOOCV, the resulting simulated SI values could be predicted accurately, with an LCCC of 0.95, $R^2$ of 0.92, RMSE of 0.4 and a bias of 0.0. The validation metrics for a simulated 1% increase in OC were better than those under derived from modelling under current conditions. This may be attributed to the simulated map showing a bimodal distribution of SI values, with approximately half of the study area predicted to be have SI values of ~0, and the other half predicted to have SI values of ~3. The reason for this is likely due to SI values returning to their natural, or expected values, that are primarily driven by clay content and clay type as opposed to land use and management. The change map shows the difference between the current observed SI values, and the simulated SI with an increase of 1% OC. This map reveals that the largest decreases in SI values were predicted to occur on dryland and irrigated cropping areas on L'lara and surrounds. Some of these areas were predicted to have their SI value decreased by up to 3 units. Much of the forested and pasture areas with lower current SI values were predicted to have their SI value largely unchanged by a 1% increase in OC content. The results of this analysis clearly show the benefit of increasing soil OC on SI and aggregate stability. This could encourage farmers and land managers to implement management practices that increase soil OC levels in cultivated areas, such as minimal tillage and cover cropping.



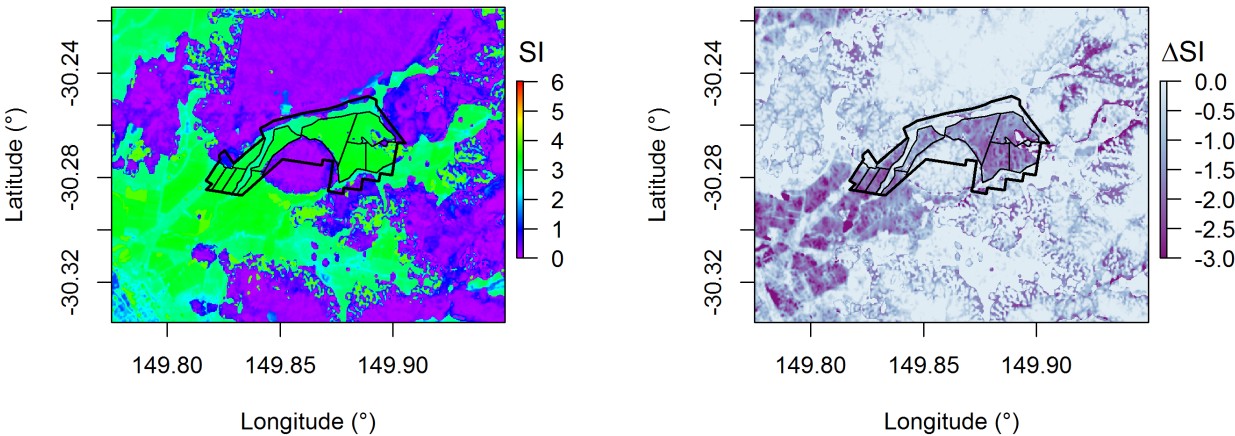

**Figure 7.** Prediction of slaking index (SI) across the study area using regression-kriging after a modelled 1% increase in organic carbon (left), and change in SI compared to current conditions (right). The boundaries of L'lara farm are indicated

## 4   Conclusions

Soil slaking index (SI) values were obtained through the use of the SLAKES smartphone application across a mixed farming landscape to assess aggregate stability of the topsoil in a mixed agricultural landscape. Land use had a clear impact on SI values, with sites under irrigated and dryland cropping showing higher SI values than those under pasture and forested areas. Clay content, CEC:clay ratio and organic carbon content had a considerable impact on SI values of soil samples. Samples with low OC and high clay content combined with high CEC:clay ratio were the most prone to slaking. An OC threshold of 1.1% was observed, below which slaking behaviour was not correlated with any of the measured soil properties and the most extreme SI values were observed. A regression-kriging approach utilising a Cubist model and diverse spatial covariates proved to be successful in spatially modelling SI. The model had high predictive power, with an LCCC of 0.85 and RMSE of 1.1, when using a LOOCV approach on the training dataset (n = 108). The results were also of high quality when assessed using an independent test set (n = 50), with an LCCC of 0.82 and RMSE of 1.1. The decrease in SI from a 1% increase in OC was also simulated and mapped across the study area. The results of this simulation suggested that considerable improvements in SI and soil aggregate stability could be achieved if practices that promote the sequestration of OC were implemented, particularly on cultivated areas. Overall, this study demonstrated that novel approaches to cheaply and rapidly assess the aggregate stability of soil samples could be combined with DSM approaches to create accurate, fine-resolution maps of aggregate stability. These maps have the potential to guide management decisions, whether that be to determine land use and management, such as avoiding cultivation/cropping in areas that are prone to slaking, or to increase OC in areas of extreme slaking through use of minimum tillage or cover-cropping.





*Author contributions.* EJ, PF and AM designed the experiment and the data analysis method. EJ, PF, RW and VP performed the field sampling campaigns. RW collected the slaking index data. EJ and PF analysed the data. EJ prepared the paper with contribution from all co-authors.

*Competing interests.* The authors declare they have no competing interests.

*Acknowledgements.* The authors would like to thank the Grains Research and Development Corporation (GRDC) and the Australian Government National Landcare Program for partly funding this research. We would also like to acknowledge the contributions of Alessandra Calegari, Vita Ayu Kusuma Dewi, Zhiwei (Vera) Wang, Bradley Ginns, Hannah Lowe and Victoria Pauly for assisting in gathering and analysing the soil data.





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
