# Peer review of "Mapping soil slaking index and assessing the impact of management in a mixed agricultural landscape"

_SOIL, 2020_

## Referee Comment (RC1) · Blandine Lemercier (Referee) · 28 Aug 2020

General comments

This paper focuses on the assessment of Slaking Index using the recent SLAKES application, and its mapping at the scale of a very diverse landscape. It is well written, pleasant to read and informative.

The study area was suitable due to the high diversity of soil and landuse situations.

This paper is original in the sense that it combine a relatively simple soil test and DSM technics to map a property of interest and produce maps useful for soil management

also being understandable for farmers. The authors even attempt to imagine and map the effect on SI of a scenario of SOC content increase. They are searching for operationality and I welcome their initiative that provides new ideas and allow progress in our community. It is daring because they based their calculation on weak relationships between SI and OC in 2 different conditions of clay content and CEC:clay ratio. Hence the use that will be made of this kind of map should be carefully managed because it could lead to simplistic interpretations and ultimately at counterproductive actions. I invite the authors to reinforce the limitation of such maps, based on a relatively limited and scattered dataset.

The effect on SI decrease of an increase of 1% OC was assessed, and the results shown a predicted map with bimodal distribution of the values. It could be interesting to test also the effect of a limited increase of 0.5%, which seems more realistic to achieve for farmers.

Specific comments

Line 22: Development of simple accessible metrics to assess soil health facilitate spatial and temporal sampling density but should also support the implication of farmers, consultants and even citizen in soil health assessment.

Line 28-29: The degree of slaking determines if the process produces a favourable or unfavourable environment for cultivation and plant growth. It is true but not sufficient. It also determines the degree of soil conservation because the aim is to cultivate but likewise to protect this resource.

Line 45: The authors focus on agricultural practices that increase soil susceptibility to slaking, but what about practices limiting susceptibility to slaking? Carbon management, crop successions, superficial or "light" tillage. . .

Line 59: Another group of widely used method to estimate aggregate stability (that is the contrary of slaking) is the Mean Weight Diameter (MWD) after wet sieving of soil

aggregates. You should mention this reference method.

Line 93-95: Please, provide the equivalent of soil references according to the World Base Reference for soil classification.

Line 96: Please define "broadacre".

Line 96: Is L'Iara covered with a soil map? If yes it and if it is relevant, it could interesting to add this map (near figure 1 for instance). It not, a land use map could also be helpful to interpret figures 6 and 7.

Line 111: "in the area surrounding L'Iara, an additional 50 samples. . ." or "50 additional samples"?

Line 108-119: Sampling scheme: collection of datasets with various sampling approach. I guess they came from various field campaigns and programmes. What are the dates for each one? A summary of the distribution of land use at the observation points is missing. It could be a table or a sentence in the text.

Line 121: What was the size of the 20 to 30 aggregates? I suppose that it was for each soil sample. Please mention that.

Line 130-132: These 2 sentence could be move to the 2.3 section and replace the 2 first sentences of this section. I suggest renaming this 2.3 section: "soil sample preparation and laboratory methods" (or something like that).

Line 141: '10 minutes'

Line 145: It the difference between replicates was more than one, only the unique additional reading was considered for the final result of SI? And what would happen if this additional reading was an outlier one? How many times a third observation was necessary?

Line 160: Please name other approaches.

Line 174: All terrain attributes are not at the same spatial resolution. Slope, aspect, MrVBF and MrRTF could have been obtained from the 5m DEM since it was available.

Line 178: Why potassium concentration is of particular interest?

Line 184: How was made the split between training and test datasets?

Line 190: "The kriged residuals was were added...". There is non information in the text about the variogram of the residuals? Were residuals spatially structured?

Line 196: The first sentence is not clear. Please reword. You could also rephrase the second sentence.

Line 198: Observation points are allocated into classes having similar behaviour. How many classes? How the choice of classes and allocations of observations was done?

Table 2: It would be relevant to distinguish training and test datasets to confirm that they cover a similar range of soil attributes values, especially because of the difference in location between the 2 datasets: training data only located within L'Iara boundaries.

Line 225: "...in these samples..." which ones? With clay content >25%?

Figure 2: It would be useful to know the number of samples in each of the classes land use/clay by adding this information in the figure. What about statistical significance of the differences between classes?

Line 267: I guess "3)" has to be suppressed.

Line 301: The scenario of an increase of SOC by 1% conduces to predict a reduction of SI of 1.59 units for soils with clay content >25% and CEC:clay ration >0.5 according to the decay function. Values of SI depending on OC are widely dispersed around the model (figure 4). Nevertheless, the map of change in SI after increase of C is based on this weak model. I suggest the authors to be more cautious in their conclusions concerning the effect of OC change on SI. Some elements of discussion about uncertainty are expected.

Line 321: "...for some models". How many models were run? Please complete the section 2.6.

Line 345: 'patterns'

Line 353-354: The accuracy of the mapping process was assessed, but not the real effect of increasing SOC content by 1% because uncertainty of the decay function of SI with SOC (the map was based on) was not estimated. This must be specified to avoid misunderstanding of this result.

---

## Referee Comment (RC2) · Sebastien Salvador-Blanes (Referee) · 5 Oct 2020

**Comments to the manuscript « Mapping soil slaking index and assessing the impact of management in a mixed agricultural landscape » submitted by Jones et al.**

This paper is a very interesting contribution to the quantification and mapping of a recently developed proxi to soil physical health. The quick and low cost soil slaking index has for the first time been successfully mapped at the landscape/farm scale, thus providing an interesting management tool to farmers through the management of the OC status of their plots. The manuscript is overall very well written. The aims, results and discussions are clearly presented, a few clarifications are needed int he methods section. I therefore recommend acceptance of the manuscript, providing a few minor issues are dealt with.

**Abstract :**

Line 13 : explain in full words the term LCCC

**Introduction :**

Lines 48-49 : state (if relevant) that an initial low soil water content increases slaking.

Lines 76-78 : add that in the paper by Annabi et al., 2017 the method used to measure soil aggregate stability is the normalized method(ISO/DIS 10930, 2012), which is time and cost consuming, which is not the case of the SLAKES approach.

**Methodology**

Lines 93-95 : please refer to the WRB soil classification as the Australian classification is unknow by mosdt of readers.

Lines 93-100 : it would be interesting to present the soil and landuse maps of the study area, as they are primary drivers of soil aggregate stability. These maps would be very useful to help the reader interprete the SI maps you present later in the paper. These data are moreover used for soil sampling as input parameters.

Lines 108-119 : the reading of this paragraph is not straightforward, as the sampling stategy is quite complex. I think the 108 samples described lines 108 to 116 should be introduced by a short sentence line 108, such as for example : "A training set of 108 samples and a test set of 50 samples were defined. The training set comprises 58 on- and 50 off-farm samples."

Lines 112-113 : why are the input parameters for the sampling strategy different for off-farm samples ? Is it due to the fact that a soil map is not available ? This could be mentioned.

Lines 113-114 : I do not understand on which sampling set the K-means clustering is applied, and for what purpose.

Line 130 : why are 20 to 30 soil aggregates necessary for the slaking test, as only 3 aggregates are necessary for the test, and the test is repeated three times at most ?

Lines 144-146 : I think it is important to provide information on the repeatability of the measurements, e.g. to ensure the average value calculated for the SI is representative of the whole sample SI. Indeed, the SI is calculated on 3 aggregates, which could be considered as a low number. It is therefore important that you provide at least a graph with the distribution of the differences in SI values for the 108 samples, including 'outlier readings'. In that respect, and to further explore the representativity of the measured aggregates, it would be interesting to present the values of the 'a' coefficient for each aggregate that is tested.

Line 145 : I do not understand what are these 'outlier readings', and on what basis they could be discarded.

Line 175 : what is the unit of the aspect ? How did you go around the circular nature of the variable ?

**Results :**

Line 209 : you state that some aggregates "increased in size by 730%". As I understand it, it is not the actual increase that is measured after 10 mn of immersion at the end of the SLAKES experiment, but rather a final aggregate size using the Gompertz function at $t=\infty$.

Line 210-211 : you mention that all SI values are below the maximum theoretical value of 7.8 suggested by Fajardo et al. (2016). What about the 'outlier readings' you mentioned line 145 ? This should be clarified.

Line 246 : just to make sure, you mention average SI values, is it an average or a median value ?

Line 261 : make reference to Table 2.

Lines 302-304 : is there a way to account for the uncertainty due to the (relatively weak) regression applied for the mapping ?

Lines 340-341 : this assumption is not straightforward, and requires to provide a soil and landuse map.

Lines 345-346 : the same deals for MrVBF : a MrVBF map would help the reader.

Lines 362-363 : I do not think the mapping of SI change is the main result that "shows" the benefit of increasing soil OC on SI values. This was shown by the results leading to Figure 4. Here, the mapping allows to precisely locate where there is a real benefit to increase soil OC to increase aggregate stability.

**Figures, tables :**

Figure 1 : the black lines (bold and not bold) on the map are not defined in the legend.

Table 3 : for readability, emphasize in bold characters the correlations that are significant at a given confidence level.

**Minor edits :**

Line 200 : "[...] on SI has been investigated"

Line 240 : "[...] natural vegetation (Fajardo et al., 2016 ; Flynn et al., 2020)."

Line 244 : remove "In a review of"

Line 267 : remove "3)"

Line 283 : remove one "been"

Line 354 : remove"under"

Line 356 : remove "be"

Line 381 : "through the use of"

---

## Author Response (AR1)

```
1
 2
    %% 2-column papers and discussion papers
 3
    \documentclass[soil, manuscript]{copernicus}
 4
    %% \usepackage commands included in the copernicus.cls:
 5
   %\usepackage[german, english]{babel}
 6
 7 %\usepackage{tabularx}
 8 %\usepackage{cancel}
   %\usepackage{multirow}
 9
10 %\usepackage{supertabular}
   %\usepackage{algorithmic}
11
12
   %\usepackage{algorithm}
13
    %\usepackage{amsthm}
14
   %\usepackage{float}
    %\usepackage{subfig}
15
16
    %\usepackage{rotating}
17
18
    \begin{document}
19
    \title{Mapping soil slaking index and assessing the impact of management in a mixed
20
        agricultural landscape}
21
    % \Author[affil]{given name}{surname}
22
23
24
    \Author[1]{Edward J.}{Jones}
25
    \Author[1]{Patrick}{Filippi}
26
    \Author[2]{Rémi}{Wittig}
27
    \Author[1]{Mario}{Fajardo}
28
    \Author[1]{Vanessa}{Pino}
29
    \Author[1]{Alex. B.}{McBratney}
30
    \affil[1]{School of Life and Environmental Sciences \& Sydney Institute of Agriculture,
31
        Faculty of Science, The University of Sydney, New South Wales, Australia}
32
    \affil[2]{École Nationale Supérieure d'Agronomie et des Industries Alimentaires (ENSAIA),
        University of Lorraine, France.}
33
34
    %% The [] brackets identify the author with the corresponding affiliation. 1, 2, 3, etc.
        should be inserted.
35
    %% If an author is deceased, please add a further affiliation and mark the respective
36
        author name(s) with a dagger, e.g. "\Author[2,$\dag$]{Anton}{Aman}" with the
        affiliations "\affil[2]{University of ...}" and "\affil[$\dag$]{deceased, 1 July 2019}"
37
38
    \correspondence{Edward J. Jones (edward.jones@sydney.edu.au)}
39
40
    \runningtitle{TEXT}
41
42
    \runningauthor{TEXT}
43
44
    \received{}
    \pubdiscuss{} %% only important for two-stage journals
45
46
    \revised{}
```

```
\accepted{}
47
48
    \published{}
49
50
    %% These dates will be inserted by Copernicus Publications during the typesetting process.
51
52
    \firstpage{1}
53
54
    \maketitle
55
56
    \begin{abstract}
57

[revised manuscript text omitted]

        Vertosols according to the Australia Soil Classification according to the World
        Reference Base for Soil Resources, with some expression of calcic horizons
        \citep{wrbisbcll}, with small areas of grey Vertosols. The sand hill area is
        represented by The soils on the sand hill area were predominately Chromosols, Dermosols
        , Kandosols, Rudosols and Tenosols \citep{isbell}Luvisol, Lixisol, Solonetz, Leptosol
        and Regosol soil groups., the unifying feature of these soils was the presence of a
        relatively sandy topsoil. L'lara encompasses a total area of 1,850 ha, with
        approximately 1,070 ha used for dryland, broadacre cropping. Cropping is performed
        primarily on the Vertipsols, and occurs over both summer and winter periods with cotton
        (\textit{Gossypium hirsutum} L.), wheat (\textit{Triticum aestivum} L.), canola
        (\textit{Brassica napus} L.) and chickpea (\textit{Cicer arietinum} L.) grown in
        rotation. Lower lying floodplain areas close to creek lines and all of the sand hill
        area is used for grazing of beef cattle on unimproved native pastures
        (\textasciitilde704 ha) and remnant forest cover (\textasciitilde76 ha).
78
79
    \begin{figure}[h]
    \includegraphics[width=120cm]{fig01.PNG}
80
    \caption{a) Location of L'lara farm and the wider study area in relation to the township of
81
        Narrabri, NSW, Australia. Sample locations used as a training set (n = 108) and test
        set (n = 50) are indicated. Satellite imagery sourced from Google Earth Pro V 7.3.2
        .5776. (March 5, 2019). Narrabri, NSW, Australia. 30° 16' 31.37"S, 149° 51' 46.42"E,
        Eve alt 20.57 km. Image © CNES/Airbus 2020. http://www.earth.google.com [April 20,
        2020]. b) MrVBF calculated at 30 m resolution using the SRTM digital elevation model. c
        ) Pixel-wise 50th percentile of NDVI calculated from Landsat 7 scenes covering the time
        -period 2000 to 2018. d) Simplified land use across the study area \citep{abares alum}.
        FR, forest reserve; Gr, grazing including understorev grazing and stock routes; DC,
        dryland cropping; IC, irrigated cropping; OW, open water; BU, built-up areas. The
        external perimeter boundary of L'lara is indicated by the thick black line and
        boundaries of cropping paddocks are indicated by thin black lines.
82
83
    \label{fig:studyArea}
84
    \end{figure}
85
86
    L'lara lies at the centre of a diverse landscape. Outside the property, dryland cropping
        and grazing occur on the floodplains and slopes to the east and south. Intensive
        irrigated agricultural production occurs on the lower floodplain to the south-west of
        the property, and the Killarney State Conservation Area lies directly to the north.
        This conservation area contains similar species as the remnant forest area found on
        L'lara which is dominated by white cypress pine (\textit{Callitris glaucophylla}),
        hickory (\textit{Acacia leiocalyx}), black cypress pine (\textit{Callitris
        endlicheri}), narrow-leaved ironbark (\textit{Eucalyptus crebra}), bulloak
        (\textit{Allocasuarina leuhmannii}) and dirty gum (\textit{Eucalyptus chloroclada}).
87
```

```
88 \subsection{Soil sampling}
```

```
89 A training set of 108 samples and a test set of 50 samples were defined (Table \ref{table
:campaigns}). The training set comprised both on- and off-farm samples. Sample sites
were identified on L'lara and the surrounding area. The majority of on-farm samples (n
```

| = 58) were identified b
type and land use as pa
of the major soil types                                                                                                                                                                                 | ased on a ra
rameters (Fi
and differe                 | andom strat
g. fi
ent land us                     | ified sampling
g:studyArea}).
es - dryland cr                                | approach utilising so
This ensured represen
opping, pasture and f                      | il
tation
orest |  |  |  |  |  |
|---------------------------------------------------------------------------------------------------------------------------------------------------------------------------------------------------------------------------------------------------------------|-------------------------------------------------------------|---------------------------------------------------------|------------------------------------------------------------------------------------|----------------------------------------------------------------------------------------------|-----------------------|--|--|--|--|--|
| cover found on the prop
surrounding L'lara, an
found within a 5 km dis
available for off-farm                                                                                                                                                        | erty. The of
additional 5
tance from t
locations t | it-farm san
50) samples
the boundar
hese sites | ples (n = <del>o inve</del>
were sourced f
y of L'lara. As
were identifie | rom neighbouring in th
rom neighbouring prop
a soil type map was
d through a random | erties
not         |  |  |  |  |  |
| <pre>stratified approach uti -resolution valley bott variables filipp</pre>                                                                                                                                                                                   | lising K-me
om flatness
iUR}. <del>K-mear</del>       | ans cluste
(MrVBF) ar
<del>IS clusteri</del>      | ering and raster
od airborne gamm
<del>ng</del> F <del>was utilis</del> e    | rs of elevation, multi
na radiometrics as inp
n <del>d to split the data i</del>       | ut
<del>nto</del>  |  |  |  |  |  |
| four strata were <del>classes</del> identified whose geographic distribution were
approximately equivalent to sand hill, transition, upper floodplain and lower
floodplain landscape positions. Sample sites were randomly selected within each stratum |                                                             |                                                         |                                                                                    |                                                                                              |                       |  |  |  |  |  |
| use not represented on L'lara. The test set was constructed utilising 30 existing
sites A supplementary dataset of 30 existing sites on the dryland cropping areas of                                                                                      |                                                             |                                                         |                                                                                    |                                                                                              |                       |  |  |  |  |  |
| also used as a test set
10 cm) sample was obtai                                                                                                                                                                                                            | for model p
ned by excav                                 | vation usir                                             | At each of the a shovel at a                                                       | e 158 sites a topsoil
discrete location.                                                  | (0 to                 |  |  |  |  |  |
| \begin{table}[h]
\caption{Summary of sampling campaigns and land use for each dataset.}
\begin{tabular}{lllccccc}                                                                                                                                       |                                                             |                                                         |                                                                                    |                                                                                              |                       |  |  |  |  |  |
| \tophline                                                                                                                                                                                                                                                     |                                                             |                                                         | 1 (=) (-) (=)                                                                      |                                                                                              |                       |  |  |  |  |  |
| & &
Sample set & Location & Da                                                                                                                                                                                                                             | &
te & Fores                                             | \multico                                                | lumn{5}{c}{Obse
e & Drvland                                                     | rvations (n)}
& Irrigated                                                                 | ~~                    |  |  |  |  |  |
| & Total \\                                                                                                                                                                                                                                                    |                                                             |                                                         |                                                                                    |                                                                                              |                       |  |  |  |  |  |
| \middlehline                                                                                                                                                                                                                                                  |                                                             |                                                         |                                                                                    |                                                                                              |                       |  |  |  |  |  |
| Training & L'lara & De                                                                                                                                                                                                                                        | c-18 & 6                                                    | & 20                                                    | & 32                                                                               | & -                                                                                          |                       |  |  |  |  |  |
| & So \\
& Surrounds & Au                                                                                                                                                                                                                                   | g-19 & 7                                                    | & 18                                                    | & 20                                                                               | & 5                                                                                          |                       |  |  |  |  |  |
| & 50 \\                                                                                                                                                                                                                                                       | 5                                                           |                                                         |                                                                                    |                                                                                              |                       |  |  |  |  |  |
| Test & L'lara & Ju                                                                                                                                                                                                                                            | 1-18 & -                                                    | & -                                                     | & 30                                                                               | & -                                                                                          |                       |  |  |  |  |  |
| & 30 \\
& L'lara & Ju                                                                                                                                                                                                                                      | 1-18 & -                                                    | \$ 20                                                   | & _                                                                                | ۶                                                                                            |                       |  |  |  |  |  |
| & 20 \\                                                                                                                                                                                                                                                       | 1 10 4                                                      | a 20                                                    | ŭ                                                                                  | u                                                                                            |                       |  |  |  |  |  |
| \bottomhline                                                                                                                                                                                                                                                  |                                                             |                                                         |                                                                                    |                                                                                              |                       |  |  |  |  |  |
| \end{tabular}                                                                                                                                                                                                                                                 |                                                             |                                                         |                                                                                    |                                                                                              |                       |  |  |  |  |  |
| <pre>table:campaigns;
\end{table}</pre>                                                                                                                                                                                                                   |                                                             |                                                         |                                                                                    |                                                                                              |                       |  |  |  |  |  |
|                                                                                                                                                                                                                                                               |                                                             |                                                         |                                                                                    |                                                                                              |                       |  |  |  |  |  |
|                                                                                                                                                                                                                                                               |                                                             |                                                         |                                                                                    |                                                                                              |                       |  |  |  |  |  |
| Sample preparat                                                                                                                                                                                                                                               | ion and l <del>t</del> at                                   | oratory me                                              | thods}                                                                             |                                                                                              |                       |  |  |  |  |  |
| All soil samples were air-d                                                                                                                                                                                                                                   | ried at 40∖t                                                | extdegree                                               | C for 48 hours.                                                                    | -A selection of 120                                                                          | to 15                 |  |  |  |  |  |
| soil <del>30</del> aggregates (Ø 5-10 mm) were isolated from the air-dried, bulk soil samples prior                                                                                                                                                           |                                                             |                                                         |                                                                                    |                                                                                              |                       |  |  |  |  |  |
| immediately evident in                                                                                                                                                                                                                                        | the hulk soi                                                | l then the                                              | sample was pag                                                                     | sed through a 5 mm si                                                                        | eve to                |  |  |  |  |  |

[revised manuscript text omitted]

```
that the footprint area has increased in size by 200\, etc. The change in SI over the
         course of the analysis is used to fit a Gompertz function on a log time-scale and
         calculate parameters \textit{a}, \textit{b}, \textit{c}:
120
     \begin{equation}
121
     SI t = ae^{-be^{-c\cdot{}log(t)}}
122
     \end{equation}
123
     where, as described in \cite{mario}, \textit{a} is an asymptote representing the maximum SI
         after an indefinite period of time, \textit{b} describes displacement along the time
         axis and is associated with initial slaking, \textit{c} describes the growth rate and
         is associated with ongoing slaking of the aggregate. The SI value returned from the
         SLAKES application is the average of the \textit{a} parameter calculated individually
         for each aggregate. A major benefit of this approach is that this value can be
         estimated after only 10 minutes of immersion, unlike othe ASWAT testr approaches that
         requires $\scaf2} hours of immersion.
124
     \subsection{Spatial covariates for modelling and mapping slaking index}
125
126
     A range of publicly available spatial datasets were used as input variables to model SI
         across the study area (Table \ref{table:covars}). This included satellite imagery, a
         digital elevation model, terrain attributes, air-borne y-radiometric maps, and a
         lithology indicator. Landsat 7 tier 1 surface reflectance satellite imagery from 2000
         to 2018 was accessed through Google Earth Engine \citep{gorelick2017google}. To remove
         pixels that were affected by cloud cover or shading, a cloud-masking filter was applied
         to all images. The Normalized Difference Vegetation Index (NDVI) was then calculated
         for each pixel in each image. The 5\textsuperscript{th}, 50\textsuperscript{th} and
         95\textsuperscript{th} percentile of the time-series of NDVI values were then
         determined for each pixel. The reason for using different NDVI percentiles was to
         characterise spatial variability in vegetation cover and vigour over the nineteen year
         period. For example, the median (50\textsuperscript{th} percentile) gives a value of
         typical greenness and the 95\textsuperscript{th} percentile gives peak plant greenness.
         The 5\textsuperscript{th} percentile would likely be low and represent soil variability
         for areas that are tilled or heavily grazed, and remain higher for areas of perennial
         cover such as forests.
127
128
     \begin{table}[h]
129
     \caption{Description and source of covariates used for digital soil mapping.}
130
     \begin{tabular}{llcl}
131
     \tophline
132
     Type
                       & Description
                                             & Resolution & Source \\
133
     \middlehline
134
     Satellite imagery$^\dag$ & Landsat 7 NDVI 5\% & 30 m
                                                                  & Google Earth Engine
                                                                                           \langle \rangle
                                                           & Google Earth Engine
135
                       & Landsat 7 NDVI 50\% & 30 m
                                                                                    11
                       & Landsat 7 NDVI 95\% & 30 m
                                                           & Google Earth Engine
136
                                                                                    11
     Terrain
                       & DEM (m)
                                                           & NSW Government
137
                                              & 5 m
                                                                            - //
138
                       & Slope (\%)
                                              & 30 m
                                                           & CSIRO \\
139
                       & Aspect (\textdegree)—
                                                           -& 30 m
                                                                         & CSIRO \\
140
                       & MrVBF
                                                           & CSIRO \\
                                              & 30 m
141
                       & MrRTE
                                              & 30 m
                                                           & CSIRO \\
                                                            & Geoscience Australia \\
142
     v-radiometrics
                       & Total dose
                                              & 100 m
143
                       & Potassium (\%)
                                              & 100 m
                                                          & Geoscience Australia \\
144
     Lithology
                       & Silica (\%)
                                              & \textasciitilde{}125 m & \cite{gray}
                                                                                        11
145
     \bottomhline
146 \end{tabular}
```

147 \belowtable{\$^\dag\$Landsat 7 NDVI values represent percentiles computed over the 2000 to 2018 time-period} % Table Footnotes

148 \label{table:covars}

149 \end{table}

[revised manuscript text omitted]

```
164
     \begin{table}[h]
     \caption{Summary statistics of slaking index and laboratory derived soil properties.}
165
166
     \begin{tabular}{lccccc}
167
     \tophline
168
     Property
                           & Min. & 1st Qu. & Median & Mean & 3rd Qu. & Max. \\
169
     \middlehline
170
     Slaking index
                                 8 0.0
                                                   & 2.6 & 2.7
                                          & 0.4
                                                                    & 4.8 & 7.3 \\
171
     Organic carbon (\%)
                                  & 0.33
                                            & 0.74
                                                      & 0.88 & 1.07
                                                                         & 1.22 & 2.97 \\
                                 & 2.5
172
     Clav (\%)
                                          & 11.1
                                                   & 29.1 & 28.1
                                                                    & 42.1 & 60.2 \\
173
     pH(1:5 H\textsubscript{2}0)
                                    & 4.8
                                              & 6.0
                                                       & 6.8 & 7.0
                                                                        & 8.3 & 9.2 \\
                                           & 0.01
174
     EC (dS m\textsuperscript{-1})
                                                     & 0.04
                                                              & 0.12 & 0.15
                                                                               & 0.19 & 0.81 \\
175
     Exch. Ca\textsuperscript{+2} (cmol\textsubscript{c} kg\textsuperscript{-1})
                                                                                   & 0.0
                                                                                             &
                & 10.5 & 11.3
                                & 19.8 & 34.0 \\
         1.7
     Exch. Mg\textsuperscript{+2} (cmol\textsubscript{c} kg\textsuperscript{-1})
176
                                                                                    & 0.0
                                                                                              &
                & 5.1 & 6.0
         0.7
                                 & 11.0 & 17.0 \\
177
     Exch. K\textsuperscript{+} (cmol\textsubscript{c} kg\textsuperscript{-1})
                                                                                 & 0.1
                                                                                           80
         .4
               & 0.8 & 0.9
                               & 1.4 & 2.2 \\
     Exch. Na\textsuperscript{+} (cmol\textsubscript{c} kg\textsuperscript{-1})
178
                                                                                            80
                                                                                  & 0.0
               & 0.2 & 0.5
                               & 0.6 & 3.6 \\
         .0
179
     CEC (cmol\textsubscript{c} kg\textsuperscript{-1})
                                                                         & 2.8
                                                                                  & 15.6 & 18.8
                                                               & 0.2
            & 32.8 & 52.8 \\
180
                     & 0.2
                               & 1.0
                                       & 1.8 & 2.9
                                                        & 3.6 & 19.4 \\
     ESP (\%)
     Ca:Mg ratio
                       & 0.1
                                 & 1.5
                                         & 1.9 & 2.2
                                                          & 2.5 & 10.8 \\
181
     CEC:clay ratio
                        & 0.01
                                  & 0.08
                                          & 0.44 & 0.43
                                                           & 0.70 & 1.09 \\
182
     \bottomhline
183
184
     \end{tabular}
     \label{table:lab_data}
185
186
     \end{table}
187
188
     Slaking index (SI) was positively correlated with clay content (r = 0.84), pH (r = 0.70),
         electrical conductivity (r = 0.44), CEC (r = 0.87), CEC:clay ratio (r = 0.84) and all
```

exchangeable cations (Table \ref{table:lab cor}). Weak negative correlations were observed for SI with OC (r = -0.31) and Ca:Mg (r = -0.26). These observations support \cite{mario} findings that SI was positively correlated with pH, clay content and exchangeable Na\textsuperscript{+} and Mg\textsuperscript{+2}, and negatively correlated with Ca:Mg. The strongest correlation with SI in this study was observed with exchangeable Mg\textsuperscript $\{+2\}$  (r = 0.90). This is in contrast to a recent study that demonstrated exchangeable Mg\textsuperscript{+2} played a negligible role in flocculation of soil particles and aggregate stability \citep{ZHU2019422}. It is believed that the observed correlation in our study is due to the dependence of exchangeable Mg\textsuperscript{+2} on clay content, CEC and shrink-swell minerals, such as smectite, rather than a direct causal effect. Clay content was a strong indicator of SI potential. Only one sample with clay content <25\% had an observed SI greater than 1, in contrast only three samples with clay content  $\gg{25}\$  had an observed SI less than 1. Clay soils are often more susceptible to slaking as they have both a higher concentration of shrink-swell minerals and also a greater concentration of smaller pores that may trap and compress air bubbles \citep{emerson64}. The majority of clay soils had a high<del>high</del> CEC:clay ratio<del>s observed in these samples and correlation</del> with clay content indicatinge that the dominant phyllosilicate in many of the clay soils studied is smectite. No correlation was observed between SI and ESP in our study. \cite{churchman} reviewed causes of swelling and dispersion in Australian soils and identified that exchangeable Na\textsuperscript{+} increased swelling, but only for high ESP values. Most of the samples in our study had low ESP values, which explains the lack of correlation with SI values. The low ESP values resulted in minimal dispersion observed in these samples, which was beneficial for this study as the SLAKES application currently cannot distinguish between slaking and dispersion \citep{mario}.

189

**190 \begin{table}[h]**

191 \caption{Pearson correlation coefficient (r) between soil properties.}
192 \begin{tabular}{corrected}

|     | (DeBruf con |      |           |       |       |        |       |           |        |                          |      |             |       |    |
|-----|-------------|------|-----------|-------|-------|--------|-------|-----------|--------|--------------------------|------|-------------|-------|----|
| 193 | OC          | &    | -0.31 &   |       | &     |        | &     | &         | &      | &                        |      | &           |       | &  |
|     | 8           | 2    | &         | &     |       | 11     |       |           |        |                          |      |             |       |    |
| 194 | Clay        | &    | 0         | 0.84} | & -0  | ).13 8 | i i   | &         | &      | &                        |      | &           | &     |    |
|     |             | &    | &         | _&    |       | &      |       | 11        |        |                          |      |             |       |    |
| 195 | рН          | &    | 0         | 0.70} | & -0  | 0.20 8 | \tex  | tbf{0.85} | &      | &                        |      | &           | &     |    |
|     | &           |      | &         | _     | &     | &      |       | &         | //     |                          |      |             |       |    |
| 196 | EC          | &    | @         | 0.44} | & 0.  | 07 8   | \tex  | tbf{0.58} | &      | \textbf{0.47}            | &    |             | &     |    |
|     | &           |      | &         | _     | &     | 8      | ۰     | &         | &      | 11                       |      |             | _     |    |
| 197 | Exch. Ca    | &    | @         | 0.83} | & -0  | 0.15 8 | \tex  | tbf{0.84} | &      | \textbf{0.83}            | &    |             | 0.45} |    |
|     | &           |      | &         | &     |       |        | &     | &         | &      | & \                      | 1    |             |       |    |
| 198 | Exch. Mg    | &    | @         | 0.90} | & -0  | .22 8  | \tex  | tbf{0.85} | &      | \textbf{0.74}            | &    | <pre></pre> | 0.45} |    |
|     | & \te       | extb | of{0.92}  | &     |       | &      |       | &         | &      | &                        | &    | 11          |       |    |
| 199 | Exch. K     | &    | @         | 0.59} | & 0.  | 19 8   | \tex  | tbf{0.65} | &      | \textbf{0.60}            | &    | <pre></pre> | 0.63} |    |
|     | & \te       | extb | of{0.67}  | &     | \text | :bf{0. | 65}   | &         | &      | &                        | &    | &           | _`    | 11 |
| 200 | Exch. Na    | &    | @         | 0.64} | & -0  | .25 8  | \tex  | tbf{0.68} | &      | \textbf{0.65}            | &    |             | 0.63} |    |
|     | & \te       | extb | of{0.60}  | &     | \text | :bf{0. | 65}   | & \text   | bf{0.5 | 2} &                     | &    | &           |       | &  |
|     | ,           | 1    |           |       |       |        |       |           |        |                          |      |             |       |    |
| 201 | CEC         | &    | @         | 0.87} | & -0  | 0.17 8 | \tex  | tbf{0.87} | &      | <pre>\textbf{0.82}</pre> | &    | <pre></pre> | 0.49} |    |
|     | & \te       | extb | of{0.99}  | &     | \text | :bf{0. | 97}   | & \text   | bf{0.7 | 0} & \tex                | tbf{ | 0.66} &     |       | &  |
|     |             | &    | 11        |       |       |        |       |           |        |                          |      |             |       |    |
| 202 | ESP         | &    | 0.01 & 0  | 0.04  | & 0.0 | )1     | & -0. | 10 & 0.16 | 8      | -0.09 & -                | 0.04 | & -0.       | 10    | &  |
|     | \text       | of{0 | ).40} & - | 0.06  | &     | &      |       | 11        |        |                          |      |             |       |    |

```
& -0.26 & 0.22 & -0.20 & -0.04 & -0.07
203
                                                             & -0.10
                                                                        & -0.27 & -0.04
     Ca:Mg
         -0.18 & -0.16 & -0.24 &
                                       \mathbf{V}
     CEC:clay & \textbf{0.84} & -0.21 & \textbf{0.74} & \textbf{0.72} & \textbf{0.38}
204
          & \textbf{0.94}
                              & \textbf{0.91}
                                                 & \textbf{0.62} & \textbf{0.53} &
         \textbf{0.94} & -0.14 & -0.16 \\
205

[revised manuscript text omitted]

270 1,00

- 280 281
- 282 \competinginterests{The authors declare they have no competing interests.} %% this section is mandatory even if you declare that no competing interests are present

```
283
```

```
284 \begin{acknowledgements}
```

285 The authors would like to thank the Grains Research and Development Corporation (GRDC) and the Australian Government's National Landcare Program for partly funding this research. The authors would also like to thank Ms Blandine Lemercier and A/prof. Sébastien Salvador-Blanes for their review and valuable suggestions to improve the manuscript, and We would also like to acknowledge the contributions of Ms Alessandra Calegari, Ms Vita Ayu Kusuma Dewi, Ms Zhiwei (Vera) Wang, Mr Bradley Ginns, Ms Hannah Lowe and Ms Victoria Pauly for their assistanceassisting in gathering— and analysing the soil samplesdate.
286 Nerd(acknowledgements)

```
286 \end{acknowledgements}
287
288
289
290 %% REFERENCES
291
292
293 \bibliographystyle{copernicus}
294 \bibliography{example.bib}
295
```

```
296 %%
```

**Author's response to RC1**

**Specific comments**

Line 22: Development of simple accessible metrics to assess soil health facilitate spatial and temporal sampling density but should also support the implication of farmers, consultants and even citizen in soil health assessment.

Sentence has been reworded to incorporate suggestions.

Line 28-29: The degree of slaking determines if the process produces a favourable or unfavourable environment for cultivation and plant growth. It is true but not sufficient. It also determines the degree of soil conservation because the aim is to cultivate but likewise to protect this resource.

Added "and has implications for soil conservation."

Line 45: The authors focus on agricultural practices that increase soil susceptibility to slaking, but what about practices limiting susceptibility to slaking? Carbon management, crop successions, superficial or "light" tillage...

Added "Techniques that increase soil organic matter such as cover-cropping, reduced tillage and application of organic amendments may reduce susceptibility to slaking"

Line 59: Another group of widely used method to estimate aggregate stability (that is the contrary of slaking) is the Mean Weight Diameter (MWD) after wet sieving of soil aggregates. You should mention this reference method.

Added "Established methods to quantify stability of aggregates subject to wet-sieving (Yoder, 1936) or simulated rainfall (Schindelbeck et al., 2016) are also time-consuming and require specialist equipment."

Line 93-95: Please, provide the equivalent of soil references according to the World Base Reference for soil classification.

Australian Soil Classification has been removed and text changed to: The soils of the floodplain area at L'lara are classified as Vertisols according to the World Reference Base for Soil Resources, with some expression of calcic horizons (IUSS Working Group WRB, 2015). The sand hill area is represented by Luvisol, Lixisol, Solonetz, Leptosol and Regosol soil groups.

Line 96: Please define "broadacre".

The term 'broadacre' has been removed to prevent any ambiguity as the term has limited use outside of Australia. The remaining sentence still conveys the same meaning.

Line 96: Is L'Iara covered with a soil map? If yes it and if it is relevant, it could interesting to add this map (near figure 1 for instance). It not, a land use map could also be helpful to interpret figures 6 and 7.

A soil type map was produced but unfortunately only covers L'lara and it is in the Australian Soil Classification and we have been requested to use WRB. MrVBF, NDVI and land use maps have been added to Figure 1. The MrVBF map gives a good indication of the distribution of Vertisols versus other soil with a sandy topsoil.

Line 111: "in the area surrounding L'Iara, an additional 50 samples. . ." or "50 additional samples"?

Text has been modified at request of RC2.

Line 108-119: Sampling scheme: collection of datasets with various sampling approach. I guess they came from various field campaigns and programmes. What are the dates for each one? A summary of the distribution of land use at the observation points is missing. It could be a table or a sentence in the text.

Text has been modified for clarity and a table added summarising the sampling dates and number of observations for each land use for each campaign.

Line 121: What was the size of the 20 to 30 aggregates? I suppose that it was for each soil sample. Please mention that.

Target diameter of "(ø 5-10 mm)" given in text.

Line 130-132: These 2 sentence could be move to the 2.3 section and replace the 2 first sentences of this section. I suggest renaming this 2.3 section: "soil sample preparation and laboratory methods" (or something like that).

Lines 130-132 moved to section 2.3 and section 2.3 renamed "Sample preparation and laboratory methods". Note number of samples changed to "12 to 15" at the request of RC2.

Line 141: '10 minutes'

Existing grammar is correct, suggestion not incorporated.

Line 145: It the difference between replicates was more than one, only the unique additional reading was considered for the final result of SI? And what would happen if this additional reading was an outlier one? How many times a third observation was necessary?

Modified text: "An additional reading was required for approximately 20\% of samples and was more commonly required for soils with higher slaking index values compared to samples which exhibited minimal slaking. When additional readings were taken the outlier reading was discarded and remaining readings averaged to provide the final SI for each sample."

When an additional reading was taken it was always within one unit of one of the original duplicates. The additional sample and the duplicate within one unit were then averaged to give the final slaking index value and the other duplicate was treated as an outlier and not used in the calculation.

Line 160: Please name other approaches.

Text has been modified "unlike the ASWAT test that requires 2 hours of immersion."

Line 174: All terrain attributes are not at the same spatial resolution. Slope, aspect, MrVBF and MrRTF could have been obtained from the 5m DEM since it was available.

The 5 m photogrammetry DEM provides the most accurate point estimate of elevation but it is not hydrologically enforced and for this reason we prefer to use the elevation derivatives calculated from the 30 m SRTM DEM.

Added "and gives an accurate point estimate of elevation though it is not hydrologically enforced".

Line 178: Why potassium concentration is of particular interest?

Added "Variation in the concentrations of the radioelements are indicative of change in soil type or parent material".

Line 184: How was made the split between training and test datasets?

Text has been updated in section 2.2. to clarify this.

Line 190: "The kriged residuals was were added. . . ". There is non information in the text about the variogram of the residuals? Were residuals spatially structured?

Added information about kriging of the residuals: "The residuals (difference between the observed and predicted SI values) at observation points showed a weak spatial autocorrelation. A Gaussian

function fit to the empirical semivariogram had a relatively large nugget of 0.81, sill of 1.11 and a range of 1.92 km."

Line 196: The first sentence is not clear. Please reword. You could also rephrase the second sentence. Line 198: Observation points are allocated into classes having similar behaviour. How many classes? How the choice of classes and allocations of observations was done?

Start of paragraph reworded "Relationships between SI and measured soil properties were explored to identify potential contributing factors as a means to inform management practices to reduce excessive slaking. Two classes of soils were evident in the samples, soils with clay content ≥25% and CEC:clay ratio≥0.5 which consistently exhibited excessive slaking, and other soils.

Table 2: It would be relevant to distinguish training and test datasets to confirm that they cover a similar range of soil attributes values, especially because of the difference in location between the 2 datasets: training data only located within L'Iara boundaries.

As indicated in Figure 1 the test set is located entirely within L'lara and is inter-mixed with samples from the training set. In this instance I don't believe it is necessary to confirm that the samples occupy the same covariate space.

Line 225: "... in these samples..." which ones? With clay content >25%?

Text modified "The majority of clay soils had a high CEC:clay ratio indicating that the dominant phyllosilicate in the clay soils studied is smectite."

Figure 2: It would be useful to know the number of samples in each of the classes land use/clay by adding this information in the figure. What about statistical significance of the differences between classes?

The number of observations for each class and significant differences (p<0.05) between means calculated using Tukey's HSD have been added to the plot and discussed in the text.

Line 267: I guess "3)" has to be suppressed.

"3)" was an incomplete reference to "(Fig. 3)". Corrected in text.

Line 301: The scenario of an increase of SOC by 1% conduces to predict a reduction of SI of 1.59 units for soils with clay content >25% and CEC:clay ration >0.5 according to the decay function. Values of SI depending on OC are widely dispersed around the model (figure 4). Nevertheless, the map of

change in SI after increase of C is based on this weak model. I suggest the authors to be more cautious in their conclusions concerning the effect of OC change on SI. Some elements of discussion about uncertainty are expected.

**Added – "provided a weak fit to the available data"**

Additional discussion points added to section 3.2.4 - "Another contributing factor for the improved validation metrics under increased OC scenarios is due to the SI values being based on modelled data which has had all unexplained error removed. Future efforts should account for the error of the underlying regression equations and quantify the uncertainty of the resultant maps by bootstrapping and applying random error based on the the prediction variance of the underlying regression equations."

Line 321: "...for some models". How many models were run? Please complete the section 2.6.

Changed to "in the model". A single cubist model was calibrated and LOOCV used for validation.

Line 345: 'patterns'

**Changed to 'features'**

Line 353-354: The accuracy of the mapping process was assessed, but not the real effect of increasing SOC content by 1% because uncertainty of the decay function of SI with SOC (the map was based on) was not estimated. This must be specified to avoid misunderstanding of this result.

Stipulated that the validation metrics refer to the "mapping procedure" and also added "Another contributing factor for the improved validation metrics under increased OC scenarios is due to the SI values being based on modelled data from which unexplained error has been removed."

The authors would like to thank RC1 for their constructive review, we have also simulated the change in slaking index under a 0.5% increase in OC as suggested.

**Author's response to RC2**

Abstract :

Line 13 : explain in full words the term LCCC

LCCC explicitly defined as Lin's concordance correlation coefficient

Introduction :

Lines 48-49 : state (if relevant) that an initial low soil water content increases slaking.

Added "and soils of low initial water content more prone to rapid and explosive slaking"

Lines 76-78 : add that in the paper by Annabi et al., 2017 the method used to measure soil aggregate stability is the normalized method(ISO/DIS 10930, 2012), which is time and cost consuming, which is not the case of the SLAKES approach.

Added "Tools that make aggregate stability quantification accessible, such as the SLAKES application, may facilitate the production of such maps." Detractions of wet-sieving and simulated rainfall techniques were added at line 59 at the request of RC1.

**Methodology**

Lines 93-95 : please refer to the WRB soil classification as the Australian classification is unknown by most of readers.

Australian Soil Classification has been removed and text changed to: The soils of the floodplain area at L'lara are classified as Vertisols according to the World Reference Base for Soil Resources, with some expression of calcic horizons (IUSS Working Group WRB, 2015). The sand hill area is represented by Luvisol, Lixisol, Solonetz, Leptosol and Regosol soil groups.

Lines 93-100 : it would be interesting to present the soil and landuse maps of the study area, as they are primary drivers of soil aggregate stability. These maps would be very useful to help the reader interpret the SI maps you present later in the paper. These data are moreover used for soil sampling as input parameters.

A soil type map was produced but unfortunately only covers L'lara and it is in the Australian Soil Classification and we have been requested to use WRB. MrVBF, NDVI and land use maps have been added to Figure 1. The MrVBF map gives a good indication of the distribution of Vertisols versus other soil with a sandy topsoil. Lines 108-119 : the reading of this paragraph is not straightforward, as the sampling strategy is quite complex. I think the 108 samples described lines 108 to 116 should be introduced by a short sentence line 108, such as for example : "A training set of 108 samples and a test set of 50 samples were defined. The training set comprises 58 on- and 50 off-farm samples."

Text has been modified for clarity.

Lines 112-113 : why are the input parameters for the sampling strategy different for off-farm samples ? Is it due to the fact that a soil map is not available ? This could be mentioned.

Correct, the soil map was only available on-farm. Text has been adjusted accordingly.

Lines 113-114 : I do not understand on which sampling set the K-means clustering is applied, and for what purpose.

K-means was the stratification method for stratified random sampling to identify off-farm samples. Text has been updated for clarity.

Line 130 : why are 20 to 30 soil aggregates necessary for the slaking test, as only 3 aggregates are necessary for the test, and the test is repeated three times at most ?

Text has been changed to "12 to 15" aggregates. While we did not need to repeat the test more than three times, however the application did crash sometimes and the test could be compromised if a shadow was inadvertently cast over the sample while analysing so it is recommended to have some spare aggregates to run additional tests. Note this has been moved to section 2.3 at the request of RC1.

Lines 144-146 : I think it is important to provide information on the repeatability of the measurements, e.g. to ensure the average value calculated for the SI is representative of the whole sample SI. Indeed, the SI is calculated on 3 aggregates, which could be considered as a low number. It is therefore important that you provide at least a graph with the distribution of the differences in SI values for the 108 samples, including 'outlier readings'. In that respect, and to further explore the representativity of the measured aggregates, it would be interesting to present the values of the 'a' coefficient for each aggregate that is tested.

Modified text: "An additional reading was required for approximately 20\% of samples and was more commonly required for soils with higher slaking index values compared to samples which exhibited minimal slaking. When additional readings were taken the outlier reading was discarded and remaining readings averaged to provide the final SI for each sample."

The graph you mention would be great to have but unfortunately the data was collected by different people over a number of months. Some reported every scan taken including replicates and outliers

for each sample, others only the final two replicates used, and others only reported the final averaged value. I will ensure that all scans are recorded and look to include such a graph in future publications, but I am reluctant to publish the incomplete dataset here.

The version of the app used reports the slaking index for each aggregate after the 10 minute analysis time but not the 'a' coefficient for each aggregate, this may be introduced in later versions of the app though.

Line 145 : I do not understand what are these 'outlier readings', and on what basis they could be discarded.

When an additional reading was taken it was always within one unit of one of the original duplicates. The additional sample and the duplicate within one unit were then averaged to give the final slaking index value and the other duplicate was treated as an outlier and not used in the calculation.

Line 175 : what is the unit of the aspect ? How did you go around the circular nature of the variable ?

Degrees symbol added to the table. The variable was not found to be a significant predictor when left in degrees or when aspect was investigated as a cardinal direction factor.

**Results :**

Line 209 : you state that some aggregates "increased in size by 730%". As I understand it, it is not the actual increase that is measured after 10 mn of immersion at the end of the SLAKES experiment, but rather a final aggregate size using the Gompertz function at  $t=\infty$ .

Correct. Added "is projected to increase".

Line 210-211 : you mention that all SI values are below the maximum theoretical value of 7.8 suggested by Fajardo et al. (2016). What about the 'outlier readings' you mentioned line 145 ? This should be clarified.

All reasonable results were below this threshold. At times when a shadow was inadvertently cast over the petri dish values of >1,000 were reported but these were discarded.

Line 246 : just to make sure, you mention average SI values, is it an average or a median value ?

It is average value. The value returned from the app is the average of the three aggregates analysed and then we average the value from duplicate tests to achieve the final value.

Line 261 : make reference to Table 2.

Reference to Table 3 added

Lines 302-304 : is there a way to account for the uncertainty due to the (relatively weak) regression applied for the mapping ?

Points added to the discussion – "Another contributing factor for the improved validation metrics under increased OC scenarios is due to the SI values being based on modelled data from which unexplained error has been removed. Future efforts should account for the error of the underlying regression equations and quantify the uncertainty of the resultant maps by bootstrapping and applying random error based on the the prediction variance of the underlying regression equations."

Lines 340-341 : this assumption is not straightforward, and requires to provide a soil and landuse map.

Land use, MrVBF and NDVI maps have been added to Figure 1 to facilitate interpretation.

Lines 345-346 : the same deals for MrVBF : a MrVBF map would help the reader.

Land use, MrVBF and NDVI maps have been added to Figure 1 to facilitate interpretation.

Lines 362-363 : I do not think the mapping of SI change is the main result that "shows" the benefit of increasing soil OC on SI values. This was shown by the results leading to Figure 4. Here, the mapping allows to precisely locate where there is a real benefit to increase soil OC to increase aggregate stability.

Correct. Sentence changed to "The produced maps highlight areas that are expected to have lower SI when OC levels are increased".

Figures, tables :

Figure 1 : the black lines (bold and not bold) on the map are not defined in the legend.

Description of lines has been added to the Fig. 1 caption as well as for Figs 6 and 7.

Table 3 : for readability, emphasize in bold characters the correlations that are significant at a given confidence level.

Bold font has been used to indicates correlation with significance at p < 0.05 and the table caption updated accordingly.

Minor edits :

Line 200 : "[...] on SI has been investigated"

**Amended**

Line 240 : "[...] natural vegetation (Fajardo et al., 2016 ; Flynn et al., 2020)."

**Amended**

Line 244 : remove "In a review of"

**Amended**

Line 267 : remove "3)"

Amended. This was an incomplete reference to Fig. 3

Line 283 : remove one "been"

**Amended**

Line 354 : remove"under"

**Amended**

Line 356 : remove "be"

Amended

Line 381 : "through the use of"

Amended